# Low-Latency Neural LiDAR Compression with 2D Context Models

**Rui Song**[1], **Yan Wang**[2], **Tongda Xu**[2], **Zhening Liu**[1], **Zehong Lin**[3*], **Jun Zhang**[1*]

[1] The Hong Kong University of Science and Technology
[2] Institute for AI Industry Research (AIR), Tsinghua University
[3] School of Data Science, Lingnan University

`rrui.song@connect.ust.hk, wangyan@air.tsinghua.edu.cn, x.tongda@nyu.edu,`
`zhening.liu@connect.ust.hk, zehonglin@ln.edu.hk, eejzhang@ust.hk`

## Abstract

Context modeling is fundamental to LiDAR point cloud compression. Existing methods rely on computationally intensive 3D contexts, such as voxel and octree, which struggle to balance the compression efficiency and coding speed. In this work, we propose a neural LiDAR compressor based on 2D context models that simultaneously supports high-efficiency compression, fast coding, and universal geometry-intensity compression. The 2D context structure significantly reduces the coding latency. We further develop a comprehensive context model that integrates spatial latents, temporal references, and cross-modal camera context in the 2D domain to enhance the compression performance. Specifically, we first represent the point cloud as a range image and propose a multi-scale spatial context model to capture the intra-frame dependencies. Furthermore, we design an optical-flow-based temporal context model for inter-frame prediction. Moreover, we incorporate a deformable attention module and a context refinement strategy to predict LiDAR scans from camera images. In addition, we develop a backbone for joint geometry and intensity compression, which unifies the compression of both modalities while minimizing redundant computation. Experiments demonstrate significant improvements in both rate-distortion performance and coding speed. The code is available at: `https://github.com/rrui-song/RangeCM`.

## 1 Introduction

LiDAR point cloud, as an effective data structure to represent real-world scenes, has been used in a wide range of applications such as autonomous driving and robotics (Guo et al., 2020). The large volume of LiDAR data creates a strong demand for effective compression algorithms. In recent years, neural networks have significantly promoted the performance of LiDAR compression. A common approach is to predict symbols based on previously decoded contextual features. Since the bitstream length is determined by the cross-entropy between the ground truth and the estimated distribution, an accurate neural context model can effectively reduce the bitrate in lossless compression. The context structure is particularly crucial for improving the density estimation accuracy, leading to the development of various context types (Gao et al., 2025; Huang et al., 2020; Wang et al., 2022a).

Although these learning-based models have greatly improved the rate-distortion performance, the coding speed remains an issue. State-of-the-art models typically rely on an informative 3D context to capture detailed local geometric features (Wang et al., 2025a; Wang & Liu, 2022; Zhou et al., 2022). Nevertheless, the heavy computational burden of processing these 3D features results in runtimes that can reach or exceed hundreds of milliseconds, making them impractical for low-latency applications. For instance, a Velodyne HDL-64E LiDAR can generate point clouds at a rate of 10 frames per second (FPS). On the other hand, although recent works (You et al., 2025) deliver real-time coding speeds, their compression performance lags behind other state-of-the-art models. Therefore, reducing the coding latency while preserving high compression efficiency remains an open challenge. Besides, existing methods typically employ two separate deep neural networks to

---

*Corresponding Authors.

calculate the dedicated context for geometry and intensity compression (Wang et al., 2025b). We argue that a single hybrid context can be applied to effectively predict both geometry and intensity, thereby reducing redundant computation and improving the coding speed.

2D range images provide a more compact and computationally efficient representation of the LiDAR point cloud. Intuitively, 2D context models can enable faster compression by operating directly on range-view features. However, extracting features from the range view is challenging due to the lack of precise 3D local contexts (Fan et al., 2021), and naively replacing the 3D context model with a 2D backbone causes severe performance degradation. To yield a superior compression ratio, state-of-the-art range image compression methods opt to use a 3D feature extractor (Zhou et al., 2022; Wang & Liu, 2022), which in turn compromises the coding speed.

In this work, we propose **RangeCM**, a fast and efficient 2D context model for LiDAR compression. It performs probability estimation based on latent features derived from a variational auto-encoder (VAE), where transforms and context models are built by the 2D convolutional neural network (CNN). By getting rid of computationally expensive 3D operators and directly working on 2D range-view features, RangeCM achieves much faster inference speed. Meanwhile, RangeCM jointly predicts geometry and intensity by integrating the context modeling of both attributes, which avoids recomputing contexts and further accelerates the inference process.

To enhance the compression performance of the 2D context model, we propose a comprehensive spatio-temporal cross-modal context structure. We first design a multi-scale context for intra-frame prediction, which decomposes the range image into a sketch map and a detail map. The estimation of details is conditioned on the sketch, which enables a coarse-to-fine next-scale prediction strategy. For inter-frame prediction, we formulate a temporal context by warping features from the reference frame to the current frame using a range-view optical flow. Furthermore, as the RGB camera is often jointly deployed with the LiDAR sensor in autonomous driving and robotic applications (Yeong et al., 2021), we develop a cross-modal context that predicts LiDAR features based on camera images. The camera context is generated using deformable attention (Zhu et al., 2021), which adaptively projects camera features onto the range view. In addition, we employ a context refinement strategy to precisely align LiDAR and camera features under the causality constraint. By aggregating diverse spatial, temporal, and camera contexts, our 2D context model even outperforms the 3D counterparts by a large margin.

We evaluate RangeCM on the Waymo Open Dataset (Sun et al., 2020) and the SemanticKITTI benchmark (Behley et al., 2019). Experiments demonstrate that RangeCM achieves significant improvements in both rate-distortion performance and coding speed. Compared to the state-of-the-art geometry compression method (Zhou et al., 2022), RangeCM yields an average BD-Rate improvement of 14.9% and 3.5× faster speed. Meanwhile, RangeCM reduces the inference latency by more than 100× compared to the state-of-the-art intensity compression model (Wang et al., 2025b), while maintaining a comparable compression efficiency. Our key contributions are as follows:

- We develop a new paradigm for low-latency LiDAR compression, where all computations are performed in the 2D domain. The proposed framework achieves state-of-the-art rate-distortion performance and practical coding speed while supporting both geometry and intensity compression in a unified manner.

- We propose a comprehensive context model that integrates spatial, temporal, and camera features for LiDAR compression. To align these distinct features, we devise a multi-scale context model for intra-frame prediction, a flow-based model for spatio-temporal aggregation, and a deformable attention module for LiDAR-camera fusion.

- We design a joint compression backbone that predicts LiDAR geometry and intensity based on a hybrid context, which merges the context modeling of geometry and intensity to improve computational efficiency.

## 2 RELATED WORK

### 2.1 POINT CLOUD COMPRESSION

Point clouds possess geometry (i.e., point coordinates) and attribute (e.g., reflecting intensities, RGB colors, and normals) information. Specialized methods have been developed to compress these two

feature types respectively. Geometry compression methods encode the orderless point cloud as more regular data structures, such as octrees (Schnabel & Klein, 2006), voxel grids (Quach et al., 2019), and range images (Wang et al., 2022b). The MPEG Geometry-based Point Cloud Compression (G-PCC) standard (Schwarz et al., 2018) follows the octree-based pipeline, where the octree is encoded by a rule-based context model losslessly. Other octree-based approaches predict the octree symbol distribution with a learning-based context model (Huang et al., 2020; Fu et al., 2022; Song et al., 2023a; Luo et al., 2024). Voxel-based methods quantize the point cloud into discrete voxels and predict the occupancy status of each voxel grid with a multi-scale context model (Wang et al., 2025a; 2022a; Nguyen et al., 2021). Besides, range image is another memory-efficient data structure to organize the point cloud. State-of-the-art range image compression methods adopt auto-regressive (Zhou et al., 2022) or multistage (Wang & Liu, 2022) context models to encode range values.

Furthermore, geometry compression can be improved by introducing temporal references. Existing methods mostly build the temporal context by searching for K-nearest neighbors (KNN) in the reference frame (Biswas et al., 2020; Song et al., 2023b; Wang et al., 2025a; Zhou et al., 2022). The symbol distribution is then predicted based on both the spatial and temporal contexts.

Traditional attribute compression methods adopt handcrafted transforms to remove the redundancy in the signal. For example, G-PCC uses region-adaptive hierarchical transform (RAHT) (De Queiroz & Chou, 2016) and predicting transform (MPEG, 2021b) to analyze attribute features. Recently, neural networks have been introduced to develop more powerful transforms and context models (Sheng et al., 2021; Fang et al., 2022; Zhang et al., 2023; Wang et al., 2025b; Zhu et al., 2025). However, these models need to recompute contextual features for attribute prediction after geometry compression, which slows down the coding speed.

## 2.2 LiDAR-Camera Fusion

Multi-modal fusion has attracted growing interest in the point cloud compression community. Several works introduce depth images as an additional prior (Wang et al., 2024; Zheng et al., 2024). However, the depth images here are only 2D projections of the point cloud, which do not introduce additional information helpful for enhancing compression. In contrast, camera images present a more promising modality, because they provide dense semantic features that the original point cloud lacks (Liu et al., 2024b). To the best of our knowledge, there is only one existing work that attempts to utilize the camera context for point cloud compression (Lin et al., 2023). This approach first uses a depth estimation network to lift the image to 3D space, then fuses camera and octree node features to enhance octree-based point cloud compression. Nonetheless, its performance is limited by the inaccurate depth estimation and unreliable LiDAR-camera alignment, achieving only marginal improvements (e.g., around 2% bitrate reduction compared to the baseline (Fu et al., 2022) at an octree depth of 10). Therefore, how to effectively utilize the camera context remains an open question.

## 3 Preliminaries

### 3.1 Range Image

The LiDAR sensor generates a point cloud by emitting $H \times W$ laser shots along $H$ elevation angles $\boldsymbol{\theta} = \{\theta_1, \cdots, \theta_H\}$ and $W$ azimuth angles $\boldsymbol{\phi} = \{\phi_1, \cdots, \phi_W\}$. To produce an $H \times W$ range image, each point is projected to a unique pixel coordinate $(m, n)$ according to the angles $(\theta_m, \phi_n)$ of the corresponding laser beam. As shown in Fig. 1, a range image pixel records the range value $r$, reflected intensity $s$, and other optional attributes of the corresponding point. The Cartesian coordinates of the point can be losslessly recovered from the range value by:

$$x = r_{i,j} \cos\theta_i \cos\phi_j, \ y = r_{i,j} \cos\theta_i \sin\phi_j, \ z = r_{i,j} \sin\theta_i, \tag{1}$$

where $(i, j)$ denotes the coordinates of the corresponding pixel in the range image. The emission angles $\boldsymbol{\theta}$ and $\boldsymbol{\phi}$ are fixed and determined by the predefined sensor scanning pattern, which is known a priori at the receiver. Therefore, the point coordinates can be determined by the scalar range value $r$, which is more efficient than transmitting the original 3D Cartesian positions.

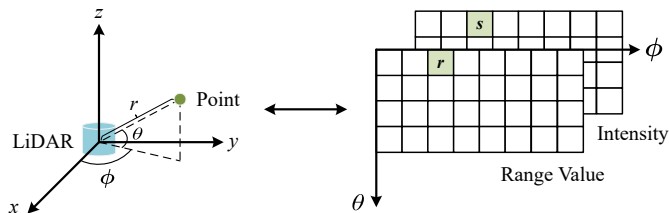

Figure 1: Illustration of range image representation of the LiDAR point cloud.

## 3.2 CONTEXTUAL VIDEO COMPRESSION

Contextual video compression employs a conditional variational auto-encoder to exploit the temporal context (Li et al., 2021a; 2024; Jia et al., 2025), which typically consists of previously decoded frames that are closely related to the current frame (Liu et al., 2024a). Given the decoded reference frame $\hat{u}$ and the current frame $x$, the model extracts an optical flow $v$ to represent the motion between the two frames. This flow is then encoded by a hyperprior-based image compression model (Ballé et al., 2018). Subsequently, the features of $\hat{u}$ are extracted and warped to the current frame using the reconstructed optical flow $\hat{v}$. The warped features serve as the temporal context $\psi_t$, which is fed into transform coding modules and the entropy model. Specifically, the analysis transform produces a latent embedding of $x$ based on $\psi_t$ as $y = g_a(x, \psi_t)$. The latent vector $y$ is quantized into $\hat{y}$ and subsequently encoded by a conditional context model formulated as:

$$p(\hat{y}|\hat{z}, \psi_t) = \prod_i (\mathcal{N}(\mu_i, \sigma_i^2) * \mathcal{U}(-0.5, 0.5))(\hat{y}_i), \tag{2}$$

$$\mu, \sigma = h_{st}(h_s(\hat{z}), h_t(\psi_t)), \tag{3}$$

where $\hat{z}$ is the quantized hyperprior encoded by a fully factorized density model $p(\hat{z})$, and $\mathcal{U}(-0.5, 0.5)$ denotes a uniform distribution centered at 0 with a width of 1. Besides, $h_s$, $h_t$, and $h_{st}$ are neural networks that predict distribution parameters based on the hyperprior and temporal context. Finally, the synthesis transform reconstructs the current frame as $\hat{x} = g_s(\hat{y}, \psi_t)$.

## 4 COMPREHENSIVE CONTEXT MODELING

### 4.1 OVERVIEW

RangeCM jointly compresses LiDAR geometry and intensity using a 2D comprehensive context. Its overall architecture is illustrated in Fig. 2. The continuous range image $x = \{r, s\}$ is quantized into $\hat{x} = \{\hat{r}, \hat{s}\}$, where $\hat{r}$ is a multi-scale representation of the range value map, and $\hat{s}$ is the quantized intensity map. In particular, $\hat{r} = \{\hat{r}_1, \hat{r}_2\}$ is given by a two-stage quantization as:

$$\hat{r}_1 = \lceil r/b_1 \rfloor, \quad \hat{r}_2 = \lceil (r - \hat{r}_1)/b_2 \rfloor. \tag{4}$$

Here, $\hat{r}_1$ is the sketched range value map, while $\hat{r}_2$ is an enhancement layer, referred to as the detail map, which conveys more details. The reconstructed range value map is recovered as $\hat{r} = \hat{r}_1 + \hat{r}_2$.

RangeCM encodes $\hat{r}_1$, $\hat{r}_2$, and $\hat{s}$ sequentially. It first adopts a deformable attention module to generate the basic camera context $\psi_c$ based on the full-precision range image $\hat{r}$. Then, a 2D CNN is employed to extract spatial features from $\hat{x}$. These features, along with the basic camera context $\psi_c$, are encoded by a VAE (Ballé et al., 2018). The spatial context $\psi_s$ is derived from the synthesis transform of the VAE, which aggregates spatial priors and the basic camera context. The temporal context $\psi_t$ is generated by a flow-based model. Subsequently, the distribution of the sketch map $\hat{r}_1$ is estimated based on both $\psi_s$ and $\psi_t$.

Although $\psi_c$ is produced by accurate geometry, fine-grained camera features may be lost during the transform coding of VAE. To overcome this issue, after recovering $\hat{r}_1$, we use another deformable attention module to compute a refined camera context $\tilde{\psi}_c$ based on the LiDAR features provided by $\hat{r}_1$. Then, the detail map $\hat{r}_2$ is predicted using a comprehensive context $\psi$, which incorporates the spatial context $\psi_s$, the temporal context $\psi_t$, and the refined camera context $\tilde{\psi}_c$. Finally, RangeCM predicts the intensity map $\hat{s}$ based on the diverse context $\psi$ and the geometric features $\hat{r}$.

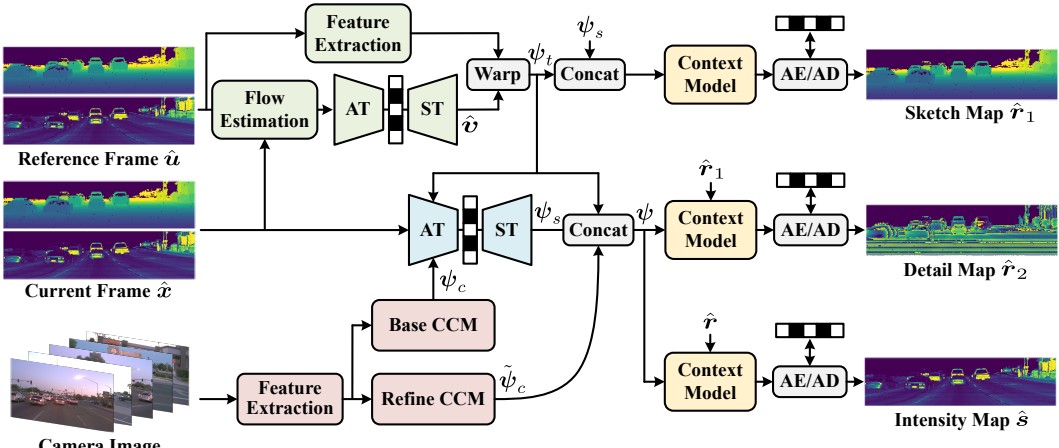

Figure 2: Architecture of RangeCM. **Blue** blocks indicate the spatial context model. **Green** blocks constitute the temporal context model. **Red** blocks represent the camera context model (CCM). AT and ST indicate analysis and synthesis transform. AE/AD stands for arithmetic encoding/decoding.

## 4.2 CAMERA CONTEXT MODEL

Practical autonomous driving and robotic systems commonly rely on the combined deployment of LiDAR and RGB cameras for robust perception. LiDAR and cameras provide complementary scene descriptions: LiDAR offers accurate geometric features, while cameras capture dense semantic information. These semantics are informative for range value prediction as well. For example, points from the same semantic instance generally have similar range values. It may be difficult to distinguish whether two points belong to the same instance from the sparse point cloud, while the camera images provide critical disambiguation. In this work, we assume that camera images are separately encoded by another image codec and that they have been decoded before LiDAR compression.

The camera context model first utilizes 2D CNNs to extract features from the range image and the camera images, respectively. Then, it adopts deformable attention (Zhu et al., 2021) to align these two modalities, using LiDAR features as the query $Q$ and camera features as the key $K$. For a specific query token $q_n$ (which corresponds to a pixel in the range image), the deformable attention module adaptively samples $N$ key tokens and computes cross-attention as follows:

$$\tilde{q}_n = \sum_{i=1}^{M} U_i \sum_{j=1}^{N} A_{ijn} V_i^T K(p_n + \Delta P_{ijn}), \tag{5}$$

where $i$ is the index of the attention head and $j$ is the index of the sampled key. Here, $K(p_n+\Delta P_{ijn})$ represents the $j^{\text{th}}$ sampled key token in the $i^{\text{th}}$ head, where the sampling position is specified by the reference point $p_n$ and the learnable offset $\Delta P_{ijn}$. Besides, $U_i$ and $V_i^T$ are learnable weights of two linear layers, while $A$ represents the weights between the query and the sampled key tokens. Both $A$ and $\Delta P$ are predicted based on $q_n$ using linear layers. Therefore, the functionality of deformable attention is to dynamically aggregate $N$ sampled camera tokens with the aggregation weights and sampling positions determined by the LiDAR query $q_n$. We further embed the deformable attention layer into a Transformer block structure (Vaswani et al., 2017), as shown in Fig. 3.

The reference point $p_n$ is a critical parameter in deformable attention, since it directly determines the correspondence between range-view and camera-view pixels. We calculate $p_n$ using the transformation matrix between LiDAR and camera. For a range image pixel, we lift it to the 3D space using Eq. (1), and project its 3D coordinates onto the camera coordinate system to obtain $p_n$. This approach provides deformable attention with an inductive bias to aggregate features from camera image pixels that are spatially close to the queried range image pixel.

Notably, LiDAR geometry is necessary to generate queries and reference points. Therefore, we cannot simply perform deformable attention using $\hat{r}$, which is unavailable at the receiver. To maintain causality, we must transmit $\psi_c$ as side information and compute $\tilde{\psi}_c$ after the decoding of $\hat{r}_1$. On the

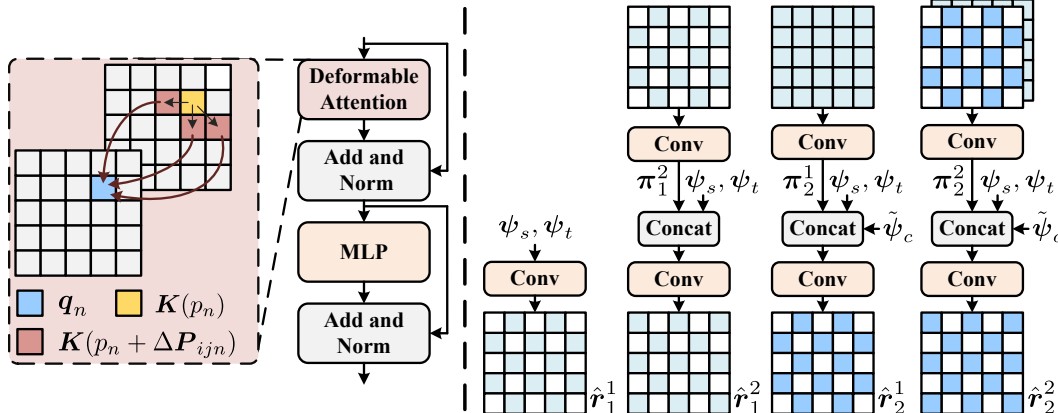

Figure 3: Left: illustration of the deformable attention block. Right: coding pipeline of the multi-scale context model.

other hand, both $\hat{r}$ and $\hat{r}_1$ preserve high-quality LiDAR geometric features, which enable accurate and effective LiDAR-camera alignment in the context model.

### 4.3 FLOW-BASED TEMPORAL CONTEXT MODEL

The temporal context model uses an optical flow to capture the accurate motion between the current and reference frames. Here, the reference frame refers to the decoded previous frame. Given the current frame $\hat{x}$ and the reference frame $\hat{u}$, it first extracts a range-view optical flow $v$ using a lightweight flow estimation model (Ranjan & Black, 2017). Since this flow is not available at the receiver, a VAE is employed to encode it as side information. Specifically, the analysis transform encodes $v$ into a latent embedding $y_v$, which is quantized into $\hat{y}_v$ and compressed based on a hyperprior $\hat{z}_v$. The synthesis transform then restores $\hat{v}$ from $\hat{y}_v$. Finally, the temporal context $\psi_t$ is produced by warping the features of $\hat{u}$ to the current view using $\hat{v}$.

### 4.4 SPATIAL PRIOR

RangeCM extracts convolutional features from $\hat{x}$ to serve as the spatial prior. This prior is then jointly encoded with the basic camera context $\psi_c$ into a latent embedding using a VAE (Ballé et al., 2018). Inspired by the contextual video compression framework (Li et al., 2021a), the transform coding is conditioned on the temporal context $\psi_t$. Specifically, we employ an analysis transform to extract the latent embedding as $y = g_a(\hat{x}, \psi_c, \psi_t)$ and generate the hyperprior as $z = h_a(y)$. Then, the quantized latent $\hat{y}$ is encoded according to Eq. (2) and Eq. (3). Finally, the synthesis transform generates the spatial context as $\psi_s = g_s(\hat{y})$.

### 4.5 MULTI-SCALE CONTEXT MODEL

We adopt a multi-scale context model to predict $\hat{r}$ in a coarse-to-fine manner, where $\hat{r}_1$ is utilized as an additional context to enhance the prediction of $\hat{r}_2$. Each map is further decomposed into two groups using a checkerboard pattern (He et al., 2021) as follows:

$$\hat{r}_1 = \left\{ \hat{r}_1^1, \hat{r}_1^2 \right\}, \quad \hat{r}_2 = \left\{ \hat{r}_2^1, \hat{r}_2^2 \right\}, \tag{6}$$

where $\hat{r}_1^1$ and $\hat{r}_2^1$ are anchors, while $\hat{r}_1^2$ and $\hat{r}_2^2$ are non-anchors. After this group partition, the context model predicts each group based on the spatio-temporal-camera context and the causal context from previous groups. The pipeline of coding $\hat{r}$ is illustrated in Fig. 3. The estimation of $\hat{r}_1$ is conditioned on the spatial context $\psi_s$, the temporal context $\psi_t$, and the causal context $\pi_1$. In contrast, $\hat{r}_2$ is predicted based on the comprehensive context $\psi$ and the causal context $\pi_2$.

As $\psi$ combines both geometry and intensity features, this hybrid context can be applied to predict both $\hat{r}_2$ and $\hat{s}$. Therefore, we adopt a lightweight prediction head to directly infer $\hat{s}$ based on $\psi$, instead of using another heavy network to recompute contextual features. For intensity compression,

Table 1: Comparison of context types, BD-Rate gains to G-PCC (%), and runtimes (in seconds). For intensity compression, we report the total runtime for coding both geometry and intensity. The best results are marked in **bold**.

| | | Geometry Compression | | | | | | | | | |
|---|---|---|---|---|---|---|---|---|---|---|---|
| | | KITTI | | | | | WOD | | | | |
| Method | Context Type | BD-Rate | Encoding | | Decoding | | BD-Rate | Encoding | | Decoding | |
| | | | Infer. | Total | Infer. | Total | | Infer. | Total | Infer. | Total |
| G-PCC | Spatial | 0 | - | 0.95 | - | 0.48 | 0 | - | 1.24 | - | 0.62 |
| EHEM | Spatial | -31.12 | 1.38 | - | 1.61 | - | - | - | - | - | - |
| RENO | Spatial | -12.47 | **0.04** | **0.07** | - | - | - | - | - | - | - |
| Unicorn | Spatio-Temp. | -27.34 | 2.65 | 2.83 | 2.36 | 2.50 | - | - | - | - | - |
| RICNet | Spatial | -45.82 | 0.40 | 0.63 | 0.40 | 0.43 | - | - | - | - | - |
| RIDDLE | Spatio-Temp. | -48.05 | - | - | - | - | -54.21 | 0.49 | 0.53 | - | 0.97 |
| RangeCM-G | Comprehensive | **-56.07** | **0.04** | 0.09 | **0.03** | **0.14** | **-61.96** | **0.14** | **0.20** | **0.09** | **0.20** |
| RangeCM-GI | Comprehensive | -51.56 | **0.04** | 0.09 | **0.03** | **0.14** | -59.94 | **0.14** | **0.20** | **0.09** | **0.20** |
| | | Intensity Compression | | | | | | | | | |
| G-PCC | Spatial | 0 | - | 0.84 | - | 0.75 | 0 | - | 0.59 | - | 0.65 |
| Unicorn | Spatial | **-12.16** | 14.84 | - | 13.04 | - | - | - | - | - | - |
| RangeCM-GI | Comprehensive | -6.96 | **0.05** | **0.10** | **0.04** | **0.17** | **-20.93** | **0.15** | **0.22** | **0.10** | **0.27** |

we first use a checkerboard pattern to decompose $\hat{s}$ into two groups $\hat{s}_1$ and $\hat{s}_2$. Then, we predict each group $\hat{s}_i$ based on the comprehensive context $\psi$, the geometry context $\hat{r}$, and the causal context $\hat{s}_{<i}$. This workflow eliminates redundant computations and significantly improves network efficiency.

### 4.6 LOSS FUNCTION

The training objective of RangeCM is to minimize the overall bitrate of encoding range values, intensity map, spatial latent, and optical flow. The corresponding loss function is:

$$\mathcal{L} = -\mathbb{E}_{\boldsymbol{x} \sim p(\boldsymbol{x})} \Big( \sum_{i=1}^{2} \log p(\hat{\boldsymbol{r}}_1^i | \boldsymbol{\pi}_1^i, \boldsymbol{\psi}_s, \boldsymbol{\psi}_t) + \sum_{i=1}^{2} \log p(\hat{\boldsymbol{r}}_2^i | \boldsymbol{\pi}_2^i, \boldsymbol{\psi}) + \sum_{i=1}^{2} \log p(\hat{\boldsymbol{s}}_i | \hat{\boldsymbol{s}}_{<i}, \hat{\boldsymbol{r}}, \boldsymbol{\psi})$$
$$+ \log p(\hat{\boldsymbol{y}} | \hat{\boldsymbol{z}}) + \log p(\hat{\boldsymbol{z}}) + \log p(\hat{\boldsymbol{y}}_v | \hat{\boldsymbol{z}}_v) + \log p(\hat{\boldsymbol{z}}_v) \Big). \tag{7}$$

We adopt the discretized Logistic mixture (Salimans et al., 2017) to fit the distribution of $\hat{r}$ and $\hat{s}$. Latent variables $\hat{y}$ and $\hat{y}_v$ are modeled by a Gaussian distribution convolved with a uniform distribution, as specified in Eq. (2). Hyperpriors $\hat{z}$ and $\hat{z}_v$ are fitted using a fully factorized density model (Ballé et al., 2018).

## 5 EXPERIMENTS

### 5.1 EXPERIMENTAL SETUP

**Datasets.** We conduct evaluations on the Waymo Open Dataset (WOD) (Sun et al., 2020) and the SemanticKITTI dataset (Behley et al., 2019). WOD provides raw range images and RGB camera images from 5 different views. It also offers accurate emission angles of LiDAR beams, which ensures lossless transformation between the range image and the point cloud. KITTI provides point cloud data along with camera images from 2 views. However, it does not provide transformation matrices between LiDAR and camera in the testing set. To strictly follow the official dataset division, we do not use camera priors for experiments on KITTI. A camera-involved RangeCM model is trained and evaluated using a different dataset partition, which is reported in Appendix D.1. Besides, KITTI provides neither range images nor beam emission angles. Following the settings in existing works (Wang & Liu, 2022; Zhou et al., 2022), our experiments are conducted on pseudo range images derived from estimated emission angles.

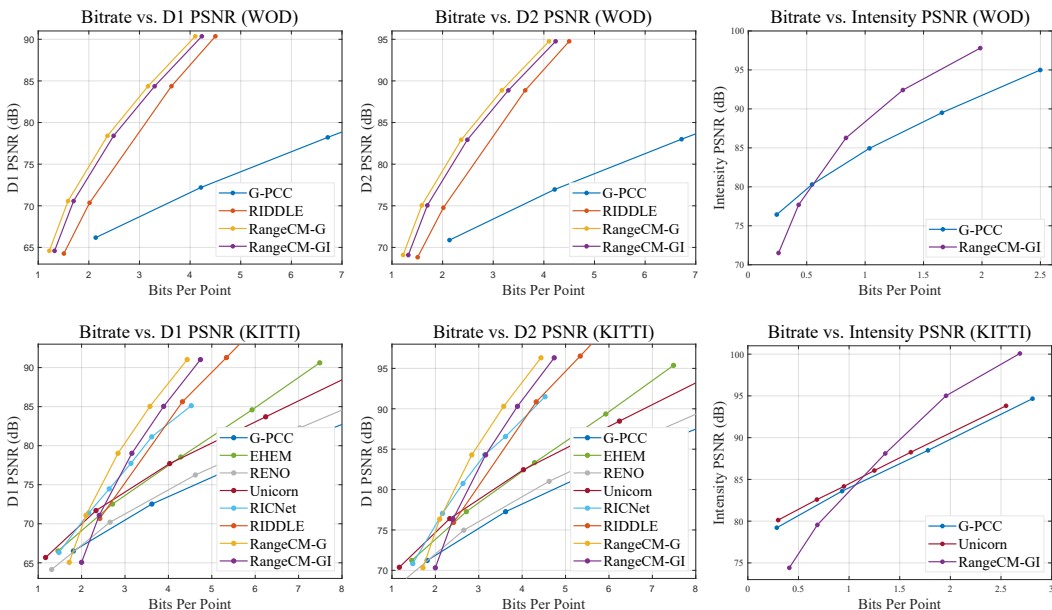

Figure 4: Rate-distortion curves on WOD and SemanticKITTI.

**Baselines.** For geometry compression, RangeCM is compared against octree-based schemes G-PCC v23 (MPEG, 2023) and EHEM (Song et al., 2023a), voxel-based models Unicorn (Wang et al., 2025a) and RENO (You et al., 2025), and range image compressors RICNet (Wang & Liu, 2022) and RIDDLE (Zhou et al., 2022). Notably, RIDDLE and Unicorn are spatio-temporal context models, while other baseline methods only exploit the spatial context. Since RIDDLE only evaluates its temporal context model on WOD, we compare its intra-frame prediction mode on KITTI instead. Meanwhile, G-PCC v23 (MPEG, 2023) and Unicorn (Wang et al., 2025b) are selected as baselines for intensity compression, where we compare RangeCM against the lossy compression modes of G-PCC and Unicorn. We also compare RangeCM with the state-of-the-art lossless LiDAR reflectance compressor SerLiC (Zhu et al., 2025) in Appendix D.6.

**Implementation Details.** For each dataset, we train two models named RangeCM-G and RangeCM-GI, respectively. RangeCM-G is exclusively optimized for geometry compression, and RangeCM-GI is trained for joint geometry-intensity compression. To avoid training multiple models for different bitrates, we randomly sample the quantization step $b_2$ during training. We evaluate RangeCM based on a single NVIDIA RTX A6000 GPU. Following the common test conditions of G-PCC (MPEG, 2021a), we adopt Point-to-Point PSNR (D1 PSNR) and Point-to-Plane PSNR (D2 PSNR) to measure the reconstruction quality. Please refer to Appendix B for more details.

## 5.2 PERFORMANCE EVALUATION

The rate-distortion performance of RangeCM is shown in Fig. 4 and Table 1. Regarding geometry compression, RangeCM outperforms existing methods by a remarkable margin. Compared to the state-of-the-art model RIDDLE, RangeCM-G and RangeCM-GI achieve the BD-rate gains of 17.14% and 12.59% on the WOD, respectively. This demonstrates the effectiveness of the proposed comprehensive context model. Besides, it is shown that octree-based and voxel-based methods (i.e., G-PCC, EHEM, and Unicorn) are more effective at low bitrates, while range image compressors (i.e., RICNet, RIDDLE, and RangeCM) perform better at high bitrates. A reasonable explanation is that octree and voxel structures can represent the point cloud with only a few symbols for coarse reconstructions at low bitrates, but the number of required symbols quickly increases as the PSNR grows, leading to inferior performance at high bitrates. In contrast, the symbol number in the range image is always fixed, thus range image compression methods are more robust to the variation of bitrate. For intensity compression, RangeCM surpasses G-PCC and achieves comparable performance to the state-of-the-art method Unicorn.

Table 2: Ablation on geometry compression.

| Model | BD-Rate to RangeCM-G |
|---|---|
| w/o CC | +6.85% |
| w/o CC and TC | +22.02% |
| w/o CC, TC, and MSC | +34.19% |

Table 3: Ablation on intensity compression.

| Model | BD-Rate to RangeCM-GI |
|---|---|
| w/o CC | +2.30% |
| w/o CC and TC | +21.88% |
| w/o CC, TC, and MSC | +31.75% |

Furthermore, RangeCM greatly reduces the coding latency compared to existing methods, which demonstrates the advantages of the proposed 2D context model. Its coding latency is around 0.1 seconds on KITTI, which satisfies the requirements of real-time applications. Compared to the real-time compressor RENO, RangeCM achieves comparable coding latency with significantly better compression efficiency. For intensity compression, RangeCM is over 100 times faster than the learning-based baseline Unicorn. Given the hybrid context, the inference latency of intensity compression is only around 10 milliseconds, because RangeCM only uses several additional layers to predict the intensity values. In contrast, Unicorn takes around 5 seconds to recompute contextual features for intensity prediction.

Since RangeCM utilizes the camera context, it requires a serial coding of camera images and Li-DAR point clouds, while other methods may process these two modalities in parallel. However, image compression can be quite fast on the GPU platform. For example, coding all 5 camera views with a GPU-accelerated JPEG codec (Nvidia, 2025c) takes only 2 milliseconds on WOD. Therefore, the serial camera-LiDAR compression of RangeCM remains much faster than the LiDAR-only compression of baseline methods.

On the other hand, RangeCM-G slightly outperforms RangeCM-GI in geometry compression, which implies that the joint geometry-intensity context modeling influences the geometry compression performance. This is probably due to the difficulty of training a versatile model. However, this performance gap is actually marginal, while the improvements on inference speed are much more significant. Thus, it is worthwhile to introduce the joint compression pipeline.

## 5.3 ABLATION STUDIES

We conduct ablation studies on WOD to validate the effectiveness of the proposed camera context model (CC), temporal context model (TC), and multi-scale context (MSC). We gradually remove these models from RangeCM-G to investigate their contributions to geometry compression. Then, we sequentially remove these models from RangeCM-GI to examine their benefits on intensity compression. The experimental results are shown in Tables 2 and 3.

The camera context model obviously benefits geometry compression, yielding a BD-Rate improvement of 6.85%. This suggests that the proposed model effectively exploits the cross-modal dependency between camera and LiDAR. However, the improvement in intensity compression is relatively modest, which is reasonable given the weak correlation between camera images and reflectance intensity. For example, the reflectance intensity is closely related to the material of real-world objects, which may be difficult to identify only from camera images. Please refer to Appendix C for detailed discussions. Meanwhile, the temporal context model significantly enhances compression performance, with BD-Rate improvements of 15.17% and 19.58% for geometry and intensity compression, respectively. Moreover, the multi-scale intra-frame context model leads to significant improvements as well.

## 6 CONCLUSION

In this work, we propose a fast and computationally efficient 2D context model for LiDAR point cloud compression. All computations are executed in the 2D domain, which yields a significantly faster inference speed compared to the 3D context models. Furthermore, the proposed method integrates the features of the current frame, reference frame, and camera images, constituting a hybrid context to facilitate effective compression. Moreover, we develop a joint geometry-intensity compression workflow by predicting both modalities based on the same hybrid context, thereby sig-

nificantly accelerating the coding process. Extensive experiments on the WOD and SemanticKITTI datasets demonstrate that the proposed universal 2D context model achieves state-of-the-art compression performance and delivers a fast coding speed that is applicable to low-latency applications.

ACKNOWLEDGMENTS

This work was supported by the General Research Fund (Project No. 16209622) from the Hong Kong Research Grants Council and by the Wuxi Research Institute of Applied Technologies, Tsinghua University under Grant 20242001120.

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

# Appendix

## A  MODEL ARCHITECTURE

**Spatial Context Extraction.**  The architecture of the VAE for spatial context extraction is shown in Fig. 5. This model aims to compress the spatial embedding of the range image along with the basic camera context. To produce the latent embedding $\boldsymbol{y}$, the $H \times W$ range image is downsampled by $4 \times 64$ times, and the derived $\boldsymbol{y}$ contains 96 channels. Distribution of $\hat{\boldsymbol{y}}$ is estimated based on the hyperprior $\hat{\boldsymbol{z}}$ and the temporal context $\boldsymbol{\psi}_t$. The dimension of the spatial context $\boldsymbol{\psi}_s$ is 64.

**Temporal Context Model.**  The temporal context model consists of a flow estimation module, a VAE for flow compression, and a motion compensation module. The flow prediction module follows the design of the spatial pyramid network (Ranjan & Black, 2017). The structure of the VAE and motion compensation module is shown in Fig. 6. The latent variable $\boldsymbol{y}_v$ and the temporal context $\boldsymbol{\psi}_t$ include 128 and 64 channels, respectively.

**Camera Context Model.**  The camera context model first employs CNN to extract features from the range image and camera images, as shown in Fig. 7. Then, it adopts two deformable attention blocks to align LiDAR and camera features. The architecture of the attention block is represented in Fig. 3 in the main paper. The deformable attention is calculated in a 128-dimensional feature space, while the camera context $\boldsymbol{\psi}_c$ contains 64 channels. The head number in deformable attention is 4, and we sample 8 key tokens in each attention head. For those range image pixels that are not projected onto any camera views, their camera features $\boldsymbol{\psi}_c$ are set to a zero tensor.

**Multi-Scale Context Model.**  After obtaining the spatial, temporal, and camera contexts, the multi-scale context model aggregates these features and predicts the distribution of $\hat{\boldsymbol{r}}_1$, $\hat{\boldsymbol{r}}_2$, and $\hat{\boldsymbol{s}}$ sequentially. The logistic mixture consists of 3 components, with each component characterized by mean, scale, and weight. Therefore, we predict 9 parameters for each symbol. The architecture of the context fusion model is shown in Fig. 8.

## B  IMPLEMENTATION DETAILS

### B.1  EXPERIMENT DETAILS

RangeCM-G and RangeCM-GI are trained using the same settings. The model is optimized for 2M and 0.7M steps on the WOD and SemanticKITTI, respectively. We use an AdamW optimizer to train the model, and the batch size is set to 8. During training, the learning rate is initially set to $1e-4$, while it decreases to $5e-5$ after $60\%$ training steps. All network modules are trained in an end-to-end manner following the loss function of Eq. 7.

For geometry compression, RangeCM uses a two-stage quantization, as defined by Eq. 4, where the quantization steps are assigned as $b_1 = 2$ and $b_2 = \{1/5, 1/10, 1/25, 1/50, 1/100\}$. For intensity compression, we adopt different settings on different datasets. The original intensity values in KITTI have been quantized into 100 levels in $[0, 1]$. We further apply another quantization with a step of $2 \times b_2$ to trade off the bitrate and reconstruction quality. For example, when the range values are quantized by $b_2 = 1/100$ in geometry compression, the intensity values are quantized with a step of $1/50$.

Intensity values in the WOD are unbounded and continuous, so we first preprocess the data as:

$$\tilde{\boldsymbol{s}} = \lceil \text{clip}(s, 0, 1) \times 255 \rceil / 255. \tag{8}$$

To adjust the bitrate and reconstruction quality, $\tilde{\boldsymbol{s}}$ is further quantized using a step of $b_2$.

Some pixels in the range image may be empty, where the laser beam does not detect any object along the corresponding direction. We represent these empty pixels with a particular symbol and encode them as ordinary pixels. In this way, the decoder can recognize these pixels.

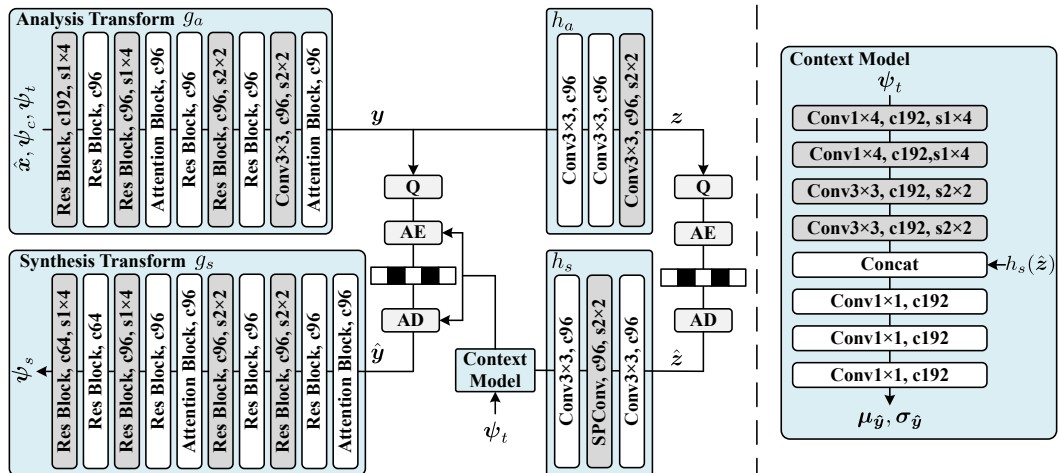

Figure 5: Architecture of the spatial context extraction model. Res Block indicates the residual block. SPConv denotes the subpixel convolution layer. Attention Block is the convolution-based attention block (Cheng et al., 2020). c represents the output dimension of the corresponding block. s indicates the stride of the downsampling or upsampling layer.

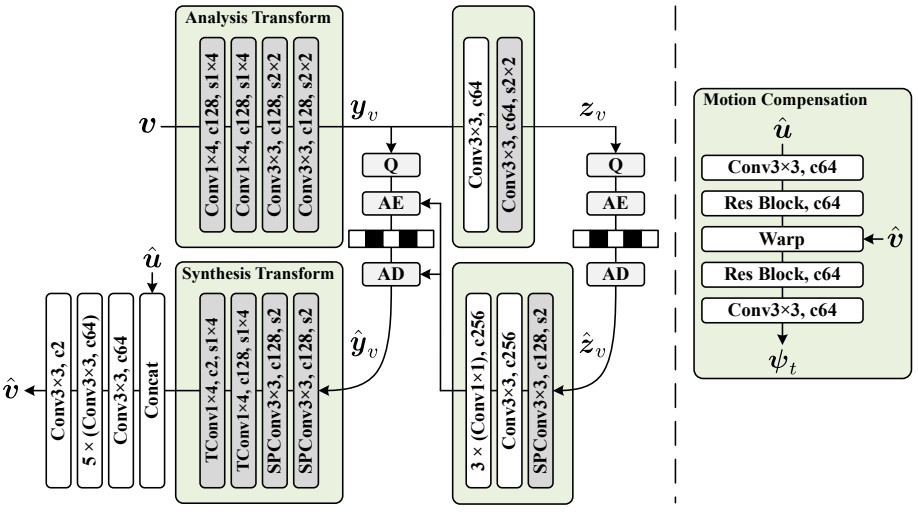

Figure 6: Left: Architecture of the optical flow compression network. TConv indicates the transpose convolution layer. Right: Structure of the motion compensation module.

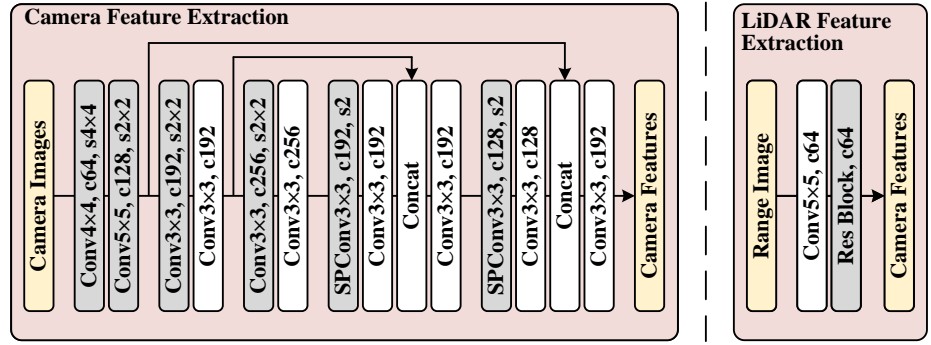

Figure 7: Architecture of the camera and LiDAR feature extraction network in the camera context model.

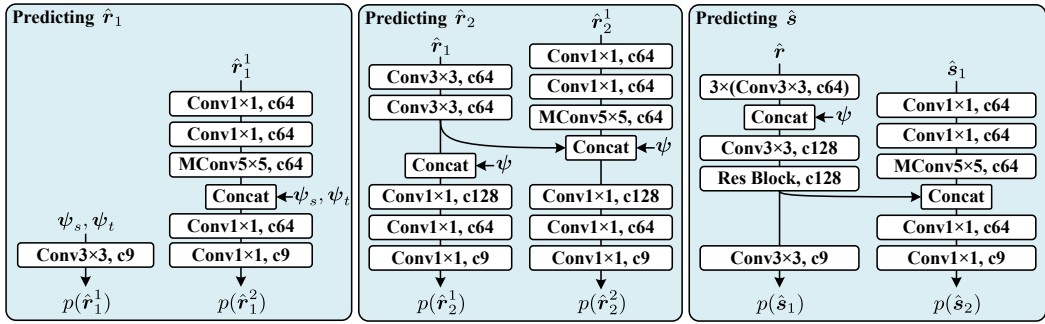

Figure 8: Architecture of the context fusion model. MConv indicates the convolution layer with a checkerboard mask (He et al., 2021).

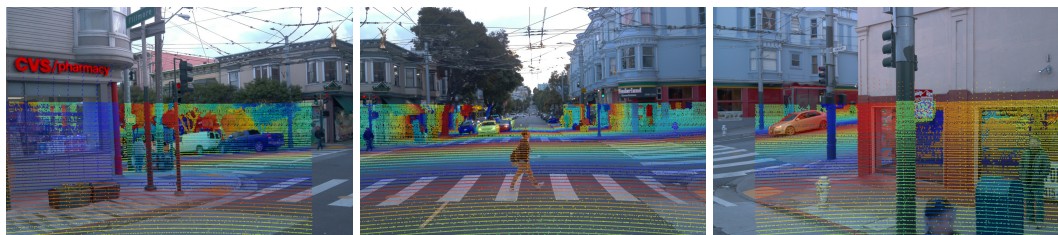

Figure 9: Visualization of the LiDAR point cloud and camera images. Points are colored according to their range value. Best viewed zoomed in.

## B.2 BASELINE SETTINGS

In geometry compression, the PSNR between the original point cloud $P$ and the reconstructed point cloud $\hat{P}$ is calculated by:

$$\text{PSNR}(P, \hat{P}) = 10 \times \log_{10}\left(\frac{3r^2}{\max\left\{\text{MSE}(P, \hat{P}), \text{MSE}(\hat{P}, P)\right\}}\right),\tag{9}$$

where $r$ is a user-specified parameter called peak value. A common practice is setting $r$ to the maximum nearest neighbor distance across the entire dataset $D$ (Biswas et al., 2020):

$$r = \max_{P \in D} \max_{p_i \in P} \min_{j \neq i} ||\boldsymbol{p}_i - \boldsymbol{p}_j||_2.\tag{10}$$

Following this formulation, we set $r$ to 57.41 and 59.70 on WOD and KITTI, respectively. While most baselines (e.g., EHEM (Song et al., 2023a) and RICNet (Wang & Liu, 2022)) use the same peak values as ours, the settings of Unicorn (Wang et al., 2025a) and RIDDLE (Zhou et al., 2022) are different. Therefore, we recompute their PSNRs under our setting to guarantee a fair comparison. Unicorn presents the MSE results, so we directly recompute the PSNR according to Eq. 9. RIDDLE releases neither source codes nor MSE results, while the chamfer distance results are provided. As the distortion of RIDDLE is completely determined by the quantization applied to the range image, we first find the quantization steps that match the reconstructed point cloud with the provided chamfer distance. Then, we use the above steps to reproduce their reconstructed point clouds, and calculate the PSNR using our peak values accordingly.

## C DISCUSSION ON CAMERA CONTEXT MODEL

LiDAR-camera fusion has emerged as an effective solution to improve the perception algorithms in autonomous driving (Li et al., 2023b; Vora et al., 2020; Liu et al., 2023). Motivated by these multi-modal perception models, the proposed comprehensive context introduces camera images as additional contexts for LiDAR compression. Although it necessitates a serial camera-LiDAR compression workflow, the image coding time is negligible using a well-optimized GPU-accelerated

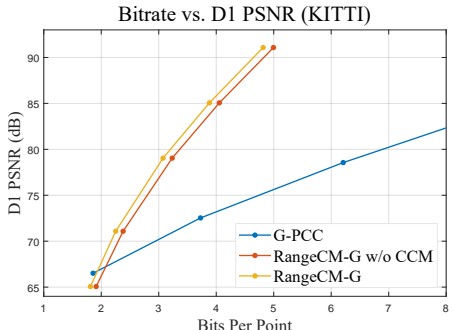 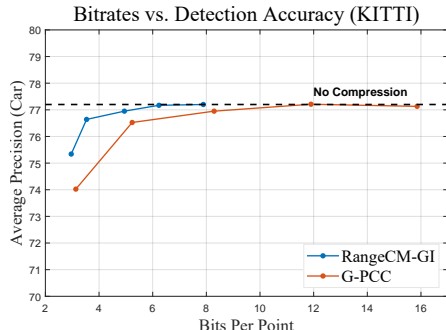

Figure 10: Left: Rate-distortion curves on the KITTI dataset. CCM means the camera context model. Right: Downstream task performance on the KITTI.

codec (Nvidia, 2025c), and we can deploy this image codec and RangeCM on the same GPU to minimize the overall coding latency. The success of these multi-modal perception methods also proves the practicality of deploying this LiDAR-camera fusion system in real-world applications.

Although most autonomous driving vehicles are equipped with both camera and LiDAR sensors, there may be special applications that require the standalone LiDAR compression (e.g., in-the-wild LiDAR compression). In this case, we can simply remove the camera context, which corresponds to the RangeCM w/o CC model in Table 2 and Table 3. Notably, this RangeCM variant maintains a BD-rate gain of 11.18% compared to the state-of-the-art method RIDDLE. Therefore, RangeCM is flexible in choosing whether to utilize the camera context, and it is not limited to applications where the camera context is available.

Besides, Table 2 and Table 3 show that the camera context is more effective on geometry compression. This is because the camera data provides dense semantic features that the sparse LiDAR point clouds lack, and these semantics are correlated to the range values, as shown in Fig. 9. However, predicting the reflectance intensity from images is challenging. Light reflectance is closely related to the materials of real-world objects. Nevertheless, it is difficult to predict the object materials from visual images. For example, wax, plastic, and crystal may appear similar in the image, while they have different reflectance properties (Li et al., 2023a). Even though the materials are known, it is still difficult to estimate the reflectance intensity without referring to the corresponding physical models. Therefore, camera context has a more significant impact on geometry compression than on intensity compression.

## D    ADDITIONAL EXPERIMENTS

### D.1    EVALUATING COMPREHENSIVE CONTEXT MODEL ON KITTI

The KITTI dataset does not provide the transformation matrices between LiDAR and camera on the testing set (i.e., sequences 11 to 21). Since all baseline methods follow the official dataset partition for training and evaluation, we exclude the camera context in our main paper to ensure the comparison is conducted on the same testing set. Here, we reorganize the KITTI dataset to evaluate the performance of the comprehensive context model. Specifically, sequences 00, 01, 02, 04, 05, 06 are selected for training, while sequences 07, 08, 09, 10 constitute the testing set. Furthermore, we train a baseline model without the camera context under this dataset division. Results in Fig. 10 show that the camera context model leads to a 4.77% BD-Rate reduction compared to the baseline.

This improvement slightly decreases compared to the one on WOD, because KITTI only includes 2 camera views, while WOD comprises 5 cameras. Consequently, fewer points can find the camera references in KITTI, thereby the benefits of the camera context are weakened. Nevertheless, modern autonomous vehicles are commonly equipped with numerous cameras (e.g., the sixth-generation Waymo Driver incorporates 13 cameras (Waymo, 2024)). As a result, most points can find the camera context in practice, leading to an effective camera context.

Table 4: Runtime (in seconds) comparison among different methods. The coding times of RangeCM are evaluated on the RTX 3080 GPU. Best results are marked in **bold**.

| | **Geometry Compression** | | | | | | | |
|---|---|---|---|---|---|---|---|---|
| | SemanticKITTI | | | | WOD | | | |
| Method | Encoding | | Decoding | | Encoding | | Decoding | |
| | Infer. | Total | Infer. | Total | Infer. | Total | Infer. | Total |
| G-PCC | - | 0.95 | - | 0.48 | - | 1.24 | - | 0.62 |
| EHEM | 1.38 | - | 1.61 | - | - | - | - | - |
| RENO | **0.04** | **0.07** | - | - | - | - | - | - |
| Unicorn | 2.65 | 2.83 | 2.36 | 2.50 | - | - | - | - |
| RICNet | 0.40 | 0.63 | 0.40 | 0.43 | - | - | - | - |
| RIDDLE | - | - | - | - | 0.49 | 0.53 | - | 0.97 |
| RangeCM-G | 0.05 | 0.09 | **0.05** | **0.09** | 0.15 | 0.21 | 0.10 | 0.23 |
| RangeCM-GI | 0.05 | 0.09 | **0.05** | **0.09** | 0.15 | 0.21 | 0.11 | 0.24 |
| | **Intensity Compression** | | | | | | | |
| G-PCC | - | 0.84 | - | 0.75 | - | 0.59 | - | 0.65 |
| Unicorn | 14.84 | - | 13.04 | - | - | - | - | - |
| RangeCM-GI | **0.06** | **0.12** | **0.04** | **0.12** | **0.16** | **0.24** | **0.12** | **0.27** |

## D.2 ABLATION STUDIES ON CONTEXT REFINEMENT

In this section, we validate the effectiveness of the context refinement strategy. Specifically, we train a RangeCM-G model without the refined camera context $\tilde{\psi}_c$ while preserving the basic camera context $\psi_c$. The context refinement strategy improves the BD-Rate by 5.11%, which verifies its importance. On the other hand, this model still outperforms the baseline without the entire camera context model (CCM), proving the benefits of the basic camera context.

## D.3 DOWNSTREAM TASK PERFORMANCE

We conduct object detection based on the decoded point clouds to investigate how compression influences the downstream task performance. Specifically, we compress both the geometry and intensity information of the point cloud, and evaluate the accuracy of the PointPillars detector (Lang et al., 2019). Results are shown in Fig. 10. Compared to G-PCC, RangeCM yields higher detection accuracy at similar bitrates.

## D.4 CODING SPEED ON ENTRY-LEVEL GPU

In the main paper, we evaluate RangeCM with an A6000 GPU. Here, we further test its coding latency with an RTX 3080 GPU. Results in Table 4 show that RangeCM still preserves faster speed than most baseline methods on this less powerful GPU.

## D.5 EVALUATION ON DIFFERENT LiDAR TYPES

In addition to 64-line LiDAR, we evaluate the performance of RangeCM on 32-line LiDAR dataset NuScenes (Caesar et al., 2020) and 128-line dataset DurLAR (Li et al., 2021b). Experimental results in Table 5 and Table 6 demonstrate that RangeCM maintains satisfactory performance on these different LiDAR types.

## D.6 LOSSLESS REFLECTANCE INTENSITY COMPRESSION

In this section, we compare RangeCM to the lossless LiDAR reflectance compressor SerLiC (Zhu et al., 2025) on KITTI and NuScenes datasets. Results in Table 7 show that RangeCM yields a comparable compression ratio to SerLiC with faster coding speed. The joint geometry-intensity compression of RangeCM remains faster than the reflectance compression of SerLiC, which demonstrates the superiority of the proposed 2D context model.

Table 5: BD-Rate improvements over G-PCC and total coding time (in seconds) on NuScenes dataset.

| Method | Geometry Compression | | | Intensity Compression | | |
|---|---|---|---|---|---|---|
| | BD-Rate | Encoding Time | Decoding Time | BD-Rate | Encoding Time | Decoding Time |
| G-PCC | 0 | 0.25 | 0.17 | 0 | 0.47 | 0.32 |
| RangeCM-GI | -37.89 | 0.08 | 0.07 | -16.89 | 0.09 | 0.08 |

Table 6: BD-Rate improvements over G-PCC and total coding time (in seconds) on DurLAR dataset.

| Method | Geometry Compression | | | Intensity Compression | | |
|---|---|---|---|---|---|---|
| | BD-Rate | Encoding Time | Decoding Time | BD-Rate | Encoding Time | Decoding Time |
| G-PCC | 0 | 1.30 | 0.64 | 0 | 1.93 | 1.33 |
| RangeCM-GI | -42.71 | 0.22 | 0.20 | -9.34 | 0.26 | 0.26 |

Table 7: Bits per point and total coding time (in seconds) for lossless reflectance compression on SemantcKITTI (left) and NuScenes (right) dataset. Please note that the coding time of RangeCM is for joint geometry-intensity compression, while the time of SerLiC is for reflectance-only compression.

| Method | BPP | Enc. Time | Dec. Time |
|---|---|---|---|
| SerLiC | 3.64 | 0.18 | 0.23 |
| RangeCM | 3.66 | 0.11 | 0.12 |

| Method | BPP | Enc. Time | Dec. Time |
|---|---|---|---|
| SerLiC | 2.52 | - | - |
| RangeCM | 2.75 | 0.09 | 0.10 |

Table 8: Bitrate distribution over different variables.

| Rate Point | $\hat{z}_v$ | $\hat{y}_v$ | $\hat{z}$ | $\hat{y}$ | $\hat{r}_1^1$ | $\hat{r}_1^2$ | $\hat{r}_2^1$ | $\hat{r}_2^2$ | $\hat{s}_1$ | $\hat{s}_2$ |
|---|---|---|---|---|---|---|---|---|---|---|
| $b_2=1/5$ | 0.57% | 1.61% | 1.24% | 9.47% | 15.09% | 12.18% | 24.68% | 18.75% | 8.84% | 7.51% |
| $b_2=1/10$ | 0.42% | 1.20% | 0.94% | 7.79% | 10.75% | 8.82% | 27.64% | 22.20% | 10.93% | 9.29% |
| $b_2=1/25$ | 0.27% | 0.76% | 0.60% | 5.89% | 6.79% | 5.59% | 29.81% | 25.12% | 13.55% | 11.57% |
| $b_2=1/50$ | 0.19% | 0.55% | 0.44% | 4.54% | 4.87% | 4.02% | 30.34% | 26.41% | 15.38% | 13.26% |
| $b_2=1/100$ | 0.14% | 0.41% | 0.33% | 3.42% | 3.62% | 2.99% | 30.13% | 27.01% | 17.04% | 14.87% |

## D.7 BIT ALLOCATION OVER VARIABLES

Table 8 presents the bitrate consumption distribution over symbol groups and latent variables. Most bits are spent on coding $\hat{r}$ and $\hat{s}$, while the latent variables $\hat{z}_v$, $\hat{y}_v$, $\hat{z}$, and $\hat{y}$ only constitute a minor proportion of the bitstream.

## D.8 ROBUSTNESS TO SCANNING PATTERN

In this section, we validate the robustness of RangeCM to different LiDAR scanning patterns (i.e., laser elevation and azimuth angles). We fine-tune RangeCM-GI on the SemanticKITTI dataset based on the model trained on WOD, as the laser emission angles are different for these two datasets. In particular, this experiment is conducted on the reorganized KITTI dataset (as introduced in Appendix D.1), because the pretrained model exploits the camera context while the testing set of the original dataset does not provide the transformation matrices between LiDAR and camera sensors. The pretrained model is fine-tuned for 9K steps. Compared to the model trained on the reorganized KITTI, the fine-tuned model yields a minor BD-Rate increase of 3.84% and 3.25% for geometry and intensity compression, respectively. The results demonstrate that the pretrained RangeCM model can be readily extended to different scanning patterns based on a simple fine-tuning process.

Table 9: Performance comparison of RangeCM-GI when compressing complete and partial LiDAR scans. For partial scan compression, we record the total bitrate for coding both left and right parts, and compute the BD-Rate against the performance of the complete scan compression. The coding time represents the runtime (in seconds) for coding complete or partial scans. Results are evaluated on the WOD.

| Data | Geometry BD-Rate | Intensity BD-Rate | Encoding Time | Decoding Time |
|---|---|---|---|---|
| Complete Scan | 0% | 0% | 0.22 | 0.27 |
| Partial Scan | +0.24% | +0.18% | 0.14 | 0.16 |

Table 10: The BD-Rate improvements (%) over baseline methods achieved by RangeCM-G and RangeCM-GI, which are evaluated on the SemanticKITTI (left) and WOD (right), respectively.

| Geometry Compression | | | | Geometry Compression | | |
|---|---|---|---|---|---|---|
| Method | RangeCM-G | RangeCM-GI | | Method | RangeCM-G | RangeCM-GI |
| G-PCC | -56.07 | -51.56 | | G-PCC | -61.96 | -59.94 |
| EHEM | -36.23 | -26.98 | | EHEM | - | - |
| RENO | -49.12 | -43.52 | | RENO | - | - |
| Unicorn | -38.78 | -32.43 | | Unicorn | - | - |
| RICNet | -12.99 | -2.90 | | RICNet | - | - |
| RIDDLE | -15.41 | -6.53 | | RIDDLE | -17.14 | -12.59 |
| Intensity Compression | | | | Intensity Compression | | |
| G-PCC | - | -6.96 | | G-PCC | - | -20.93 |
| Unicorn | - | +7.03 | | Unicorn | - | - |

### D.9 PARTIAL SCAN COMPRESSION

The common practice of LiDAR point cloud compression is to encode the complete scan. In this setting, the codec compresses the current frame $x_t$ when the sensor is scanning the next frame $x_{t+1}$, and the quantized previous frame $\hat{x}_{t-1}$ is used as the temporal context. Nevertheless, in some latency-sensitive applications, the LiDAR compressor needs to compress the partial point cloud during the scanning of the sensor. To validate the performance of RangeCM on encoding the partial scan, we split the range image vertically into two uniform parts. Then, RangeCM-GI is employed to encode the left and right parts of the image independently. The experimental results are shown in Table 9. When compressing the partial scan, RangeCM achieves almost the same performance as encoding the complete scan. Therefore, RangeCM can be directly applied to coding partial point clouds. Moreover, the coding latency for partial scan compression decreases accordingly.

### D.10 MORE BASELINES

In the main paper, RangeCM is compared against state-of-the-art LiDAR compression methods. Here, we additionally compare it with another baseline MuSCLE (Biswas et al., 2020). MuSCLE utilizes a spatio-temporal context and supports both geometry and intensity compression. However, it did not provide the PSNR results on intensity compression or the coding latency, so we only compare RangeCM with it regarding geometry compression performance. On the SemanticKITTI dataset, RangeCM-GI outperforms MuSCLE with a BD-Rate of -49.69%, which verifies the strength of RangeCM.

### D.11 BD-RATE TO BASELINES

In the main paper, we compare the BD-Rate improvements over G-PCC yielded by different methods. Here, we provide the specific BD-Rate improvements of RangeCM compared to each baseline method. The corresponding results are shown in Table 10.

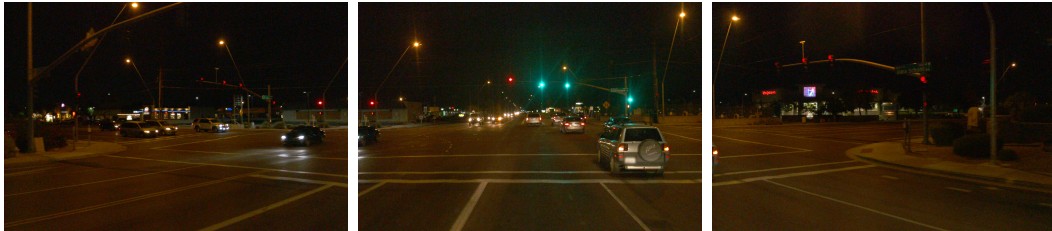

Figure 11: Visualization of camera images captured in the dark scene.

## E  VISUALIZATION

We visualize the sampling locations and attention weights predicted by the deformable attention model in Fig. 12. Specifically, these results are collected from the last attention block in the basic camera context model. It is shown that deformable attention tends to aggregate features from similar semantic objects. In addition, the sampling points close to the reference point generally have higher attention scores.

Furthermore, we visualize the reconstructed optical flow $\hat{v}$ given by the temporal context model, as shown in Fig. 13. We also present range value maps of the current and reference frame to visualize the ground-truth motion patterns. These results prove that the estimated flow effectively captures the motions between adjacent frames, which provides accurate correlations for temporal context modeling. Besides, we also visualize the original and decoded point clouds in Fig. 14.

## F  DISCUSSION ON IN-VEHICLE DEPLOYMENT

This paper evaluates RangeCM's efficiency on the general-purpose GPU such as RTX A6000 and RTX 3080. In practice, the in-vehicle GPU platform provides comparable or even stronger computing performance. For example, the Nvidia Drive AGX Thor platform offers 1000 Tera Operations Per Second (TOPS) for processing INT8 data (Nvidia, 2025b). In contrast, the RTX A6000 and RTX 3080 GPU offer only 619.4 and 476 TOPS (Nvidia, 2025a). Furthermore, the AGX Thor platform is built on the latest Blackwell architecture, while the RTX A6000 and RTX 3080 are based on the previous-generation Ampere architecture. Besides, the peak memory consumption of RangeCM is only 3GB, while AGX Thor has 64GB system memory (shared by CPU and GPU), which is sufficient to deploy our model. Finally, the AGX Thor platform adopts hardware-level optimization to accelerate the I/O of camera and LiDAR data, which enables fast I/O speed. Therefore, we believe RangeCM can preserve low-latency coding capability in practical in-vehicle deployment.

## G  LIMITATION

As RangeCM utilizes the camera images as the context, its performance may be affected by the image quality. For example, the camera images are less informative in low visibility weather conditions (e.g., fog and rain) or dark scenes (e.g., night), which in turn influences the effectiveness of the proposed method. Compared to its average performance, we found that RangeCM costs 24.4% more bitrates to encode a sequence captured in the dark environment. The images from this sequence are visualized in Fig. 11.

Furthermore, the intensity compression performance may decline when dealing with real-world objects that have complex reflectance properties. For instance, it is difficult to predict the accurate reflectance intensity from woodland, water, and snow.

## H  LLM USAGE STATEMENT

LLM is not applied for any research ideation. We only use the LLM to detect grammar errors and polish the human-written manuscript.

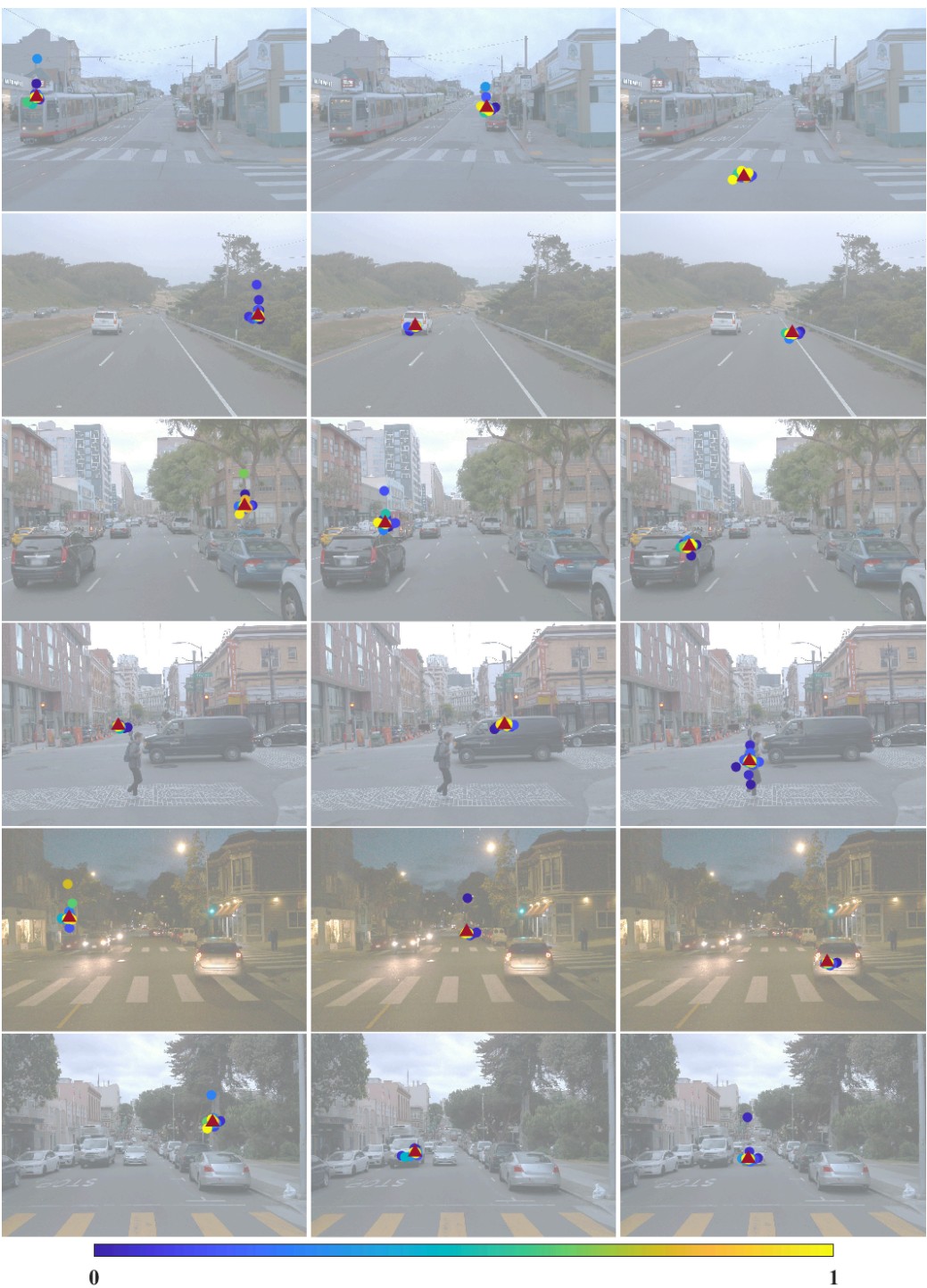

0                                                                                          1

Figure 12: Visualization of the deformable attention model. **Red** triangle indicates the reference point, i.e., the 2D projection of the query. Dots represent the predicted sampling locations, and they are colored according to their attention scores.

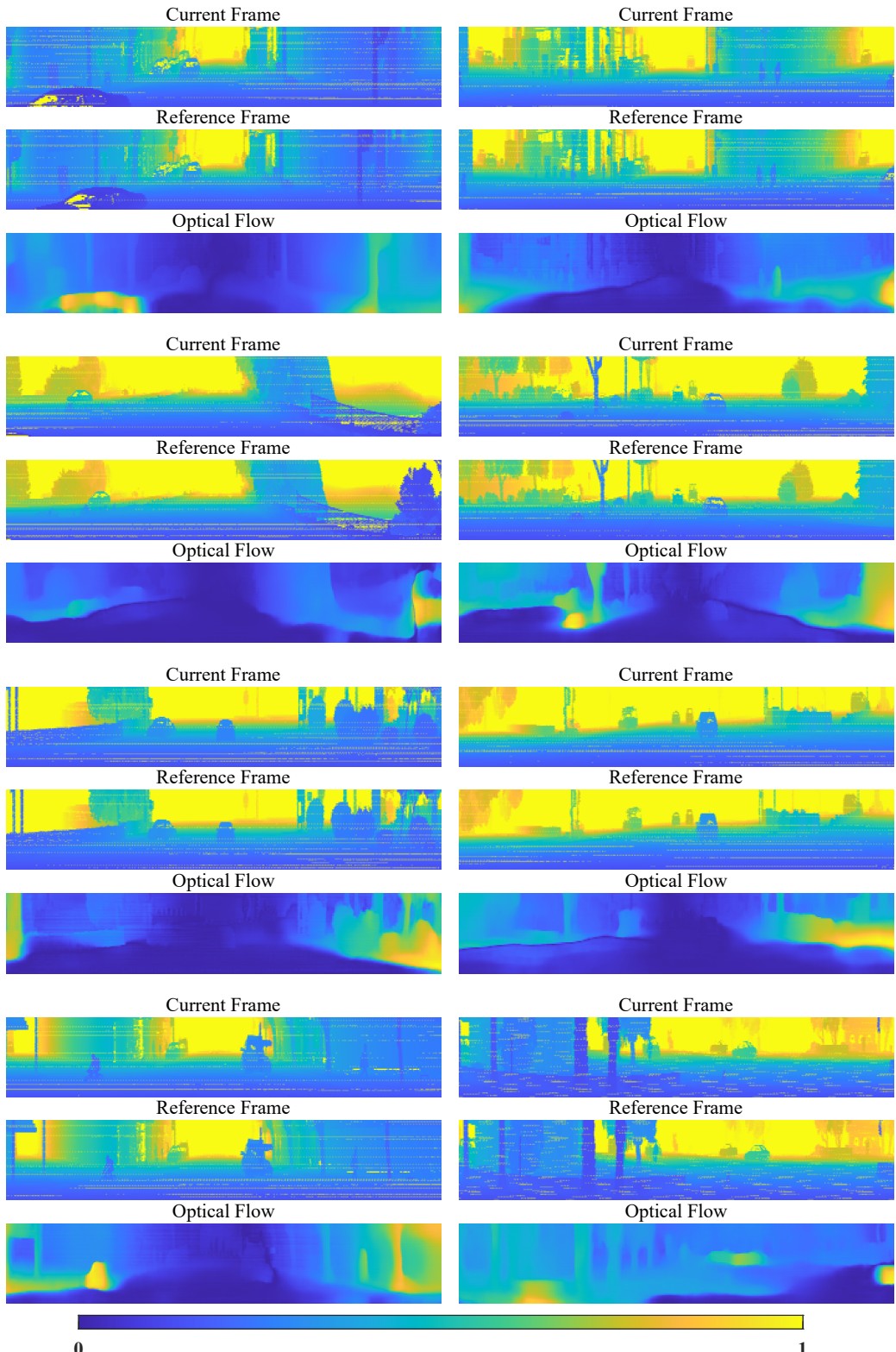

Figure 13: Visualization of the current frame, reference frame, and reconstructed optical flow.

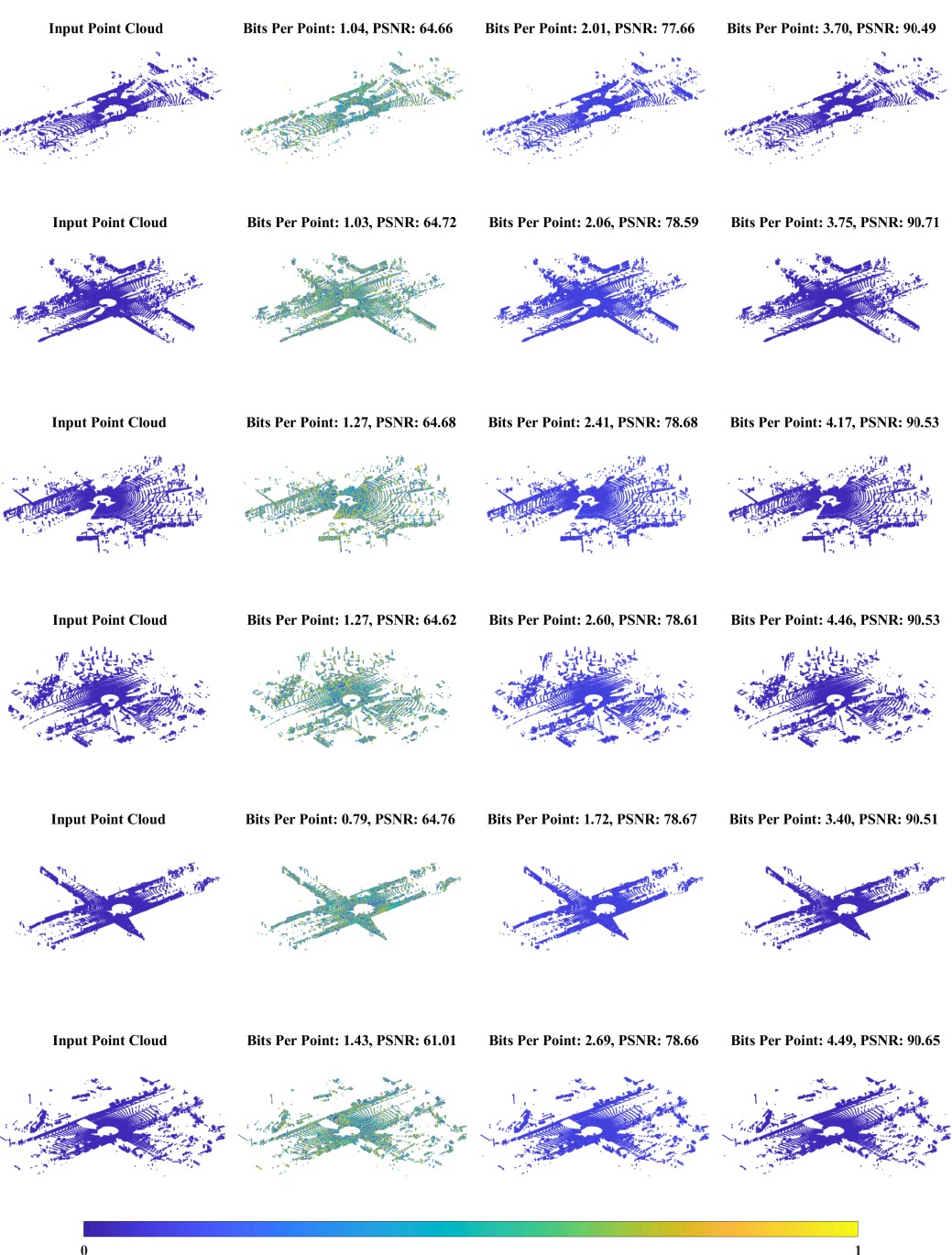

Figure 14: Visualization of the ground truth and reconstructed point clouds at different bitrates. Color indicates the chamfer distance between the ground truth and the decoded point cloud.

