# OpenReview forum: "Low-Latency Neural LiDAR Compression with 2D Context Models"
_ICLR.cc/2026/Conference — ICLR 2026 Poster_

### Official Review · Reviewer_MSXZ · 2025-10-31

**Soundness:** 3
**Presentation:** 3
**Contribution:** 3
**Rating:** 8
**Confidence:** 3

**Summary:**

The paper addresses the LiDAR point cloud compression problem via context modeling. The method yields a significant performance increase and faster runtime than its competitors. In order to achieve that,
all computations are performed in the 2D domain (images from the camera and range images from LiDAR). Furthermore, the proposed method utilizes context from camera features for LiDAR compression.
Evaluations were performed on WOD and KITTI datasets, and comparisons to prior works are properly presented.

**Strengths:**

1. The paper clearly makes and supports its claims through empirical evaluations.
2. Components of the proposed method are clearly explained.

**Weaknesses:**

1. Storage is a major point of compression. What is the effective storage saving that the proposed method offers, compared to prior works, if any?

**Questions:**

1. How would the authors address imperfect data? For example, there might be misalignments between the camera and LiDAR or fast motions that cause smeared images.
2. Why are some prior works not included in Figure 4?

---

> ### Author Response · Authors · 2025-11-20
> **Response to Reviewer MSXZ**
>
> We sincerely appreciate the reviewer for the thorough review and valuable suggestions. We hope the following responses can address your concerns.
>
> **W1: Storage savings compared to prior works**
>
> The BD-Rate improvements over baseline methods (i.e., the percentage of the storage saving delivered by RangeCM under the same reconstruction quality)  are shown in the following tables. Besides, we have included these results in **Appendix D.11**.
>
> **Saved storage (%) on the SemanticKITTI**
>
> |  Method | RangeCM-G (Geometry Compression) | RangeCM-GI (Geometry Compression) | RangeCM-GI (Intensity   Compression) |
> |:-------:|:--------------------------------:|:--------------------------------:|:------------------------------------:|
> |  G-PCC  |              -56.07              |              -51.56              |                 -6.96                |
> |   EHEM  |              -36.23              |              -26.98              |                   -                  |
> |   RENO  |              -49.12              |              -43.52              |                   -                  |
> | Unicorn |              -38.78              |              -32.43              |                 +7.03                |
> |  RICNet |              -12.99              |              -2.90               |                   -                  |
> |  RIDDLE |              -15.41              |               -6.53              |                   -                  |
>
> **Saved storage (%) on the WOD**
>
> | Method | RangeCM-G (Geometry Compression) | RangeCM-GI (Geometry Compression) | RangeCM-GI (Intensity   Compression) |
> |:------:|:--------------------------------:|:--------------------------------:|:------------------------------------:|
> |  G-PCC |              -61.96              |              -59.94              |                -20.93                |
> | RIDDLE |              -17.14              |              -12.59              |                   -                  |
>
> **Q1: Imperfect Data**
>
> As the deformable attention model has learned the accurate LiDAR-camera alignment pattern from the training on the perfectly aligned dataset, it can adaptively correct the minor to moderate misalignment between camera and LiDAR sensors by applying the learnable sampling offset $\Delta P$. If the camera and LiDAR sensor are severely misaligned, it would be better to perform accurate calibration before compressing the data. To address the smeared images caused by fast motion, we can first correct the rolling shutter effect using the algorithm proposed in [1], and then perform compression as usual.
>
> **Q2: Why are some prior works not included in Figure 4**
>
> As most baseline methods provide neither the code nor sufficient details for reproduction, we directly quote the results from the corresponding papers for comparison. The entries in Table 1 and Figure 4 are left blank if the baseline paper did not present the corresponding results. On the other hand, since RangeCM and the baselines use the same experiment settings (e.g., training-testing dataset partition, evaluation metrics), the comparison is still fair and reliable. Besides, we have updated the rate-distortion curve of RENO in Fig. 4.
>
> We are pleased to provide further explanations if you have any remaining questions.
>
> **References**
>
> [1] Scalability in Perception for Autonomous Driving: Waymo Open Dataset. CVPR 2020.

---

### Official Review · Reviewer_uEV9 · 2025-11-01

**Soundness:** 2
**Presentation:** 3
**Contribution:** 2
**Rating:** 4
**Confidence:** 4

**Summary:**

The paper proposes RangeCM, a neural LiDAR compression framework that integrates spatial, temporal, and camera contexts to achieve low-latency and high-efficiency range–intensity compression. It introduces deformable attention for LiDAR–camera alignment and a dual-stage quantization strategy for geometry encoding. Experiments on Waymo and SemanticKITTI show rate–distortion and latency improvements over G-PCC, RIDDLE, and Unicorn.

**Strengths:**

1. The paper presents a coherent end-to-end architecture integrating spatial, temporal, and camera contexts for LiDAR compression, showing a thoughtful system design.
2. It consistently outperforms prior methods like RIDDLE and Unicorn in both compression ratio and reconstruction quality, demonstrating effectiveness across datasets.

**Weaknesses:**

1. The claimed low-latency perception is not experimentally validated, as all tests rely on complete point clouds rather than partial-scan or early-sensing data.
2. The use of ground-truth references for each frame hides cumulative decoding errors, leaving long-term stability and error propagation unexamined.
3. Inference latency comparisons may be biased, since the hardware configuration of baseline methods is unspecified.
4. The absence of released models or code prevents independent verification and limits the reproducibility of results.

**Questions:**

1. The method claims early scene prediction before full scanning, yet all tests use complete point clouds. Does low latency refer only to reduced inference time rather than partial-scan prediction?
2. Since each frame uses the ground-truth reference instead of reconstructed data, has the impact of error propagation in continuous decoding been evaluated?
3. Why are recent temporal compression baselines like MuSCLE[1] and BIRD-PCC[2] omitted from comparisons or tables?
4. Were all latency comparisons conducted on the same hardware (e.g., RTX A6000)? If not, how are timing results normalized?
5. The experimental results in Table 1 contain many missing entries, and several baselines are not uniformly reproduced across datasets. Given that the paper’s main contribution lies in introducing the Camera Context, the KITTI evaluation omits this component and instead reuses baseline results due to unavailable calibration matrices. Does this inconsistent experimental design—using different model implementations for different datasets—undermine the fairness and reliability of the reported comparisons?
6. The appendix reports downstream task performance on KITTI, which is informative. However, the results suggest that G-PCC achieves nearly identical downstream accuracy when doubling data usage (6→12), while also benefiting from simple parallelization, low end-to-end latency, and broad generalization without GPU dependency. Considering modern increases in communication bandwidth, under what specific conditions does the proposed method provide a clear practical advantage over G-PCC—particularly in real-time deployment scenarios where latency and generality may outweigh compression ratio improvements?
7. The paper repeatedly claims real-time capability, yet all experiments are conducted on an RTX A6000 GPU—a configuration rarely available in real-world automotive or mobile platforms. Considering that true real-time performance requires end-to-end processing at sensor-level frame rates (≥10 FPS) under resource-constrained conditions, can the authors clarify the specific hardware assumptions and computational settings under which their method achieves real-time operation (e.g. memory, computational requirements)? If such conditions cannot be met on embedded or on-vehicle systems, should the claim be limited to low-latency rather than real-time performance?
8. Other problems.
1) [152] Equation (1) misuses angle indices: x and y depend on \( \theta_i \), but \( z\) incorrectly uses \(\theta_j\) instead of \(\theta_i\), causing an inconsistency in coordinate conversion.
2) [197] The phrase “full-precision range value map can be recovered as \hat r = \hat r_1 + \hat r_2” is misleading because both terms are quantized. It should be described as a “reconstructed” or “approximate” range map .
3) Figure 2 contains a typographical error “Intenstiy Map”.

[1] Biswas S, Liu J, Wong K, et al. Muscle: Multi sweep compression of lidar using deep entropy models[J]. Advances in Neural Information Processing Systems, 2020, 33: 22170-22181.
[2] Liu C S, Yeh J F, Hsu H, et al. Bird-pcc: Bi-directional range image-based deep lidar point cloud compression[C]//ICASSP 2023-2023 IEEE International Conference on Acoustics, Speech and Signal Processing (ICASSP). IEEE, 2023: 1-5.

---

> ### Author Response · Authors · 2025-11-20
> **Response to Reviewer uEV9 - 1**
>
> We sincerely appreciate the reviewer for the thorough review and valuable suggestions. We hope the following responses can address your concerns.
>
> **W1 and Q1: Compressing partial-scan data**
>
> RangeCM aims to reduce the coding latency rather than the perception time. We do not intend to claim the low-latency perception or the early scene prediction strategy. The “low-latency” refers to reducing the compression latency, instead of the partial-scan prediction time.
>
> Please note that the common setting in point cloud compression is to compress the complete LiDAR point cloud after scanning, rather than encoding the partial point cloud during the ongoing scanning process. For example, the compressor may encode the current frame $x_t$ when the sensor is scanning the next frame $x_{t+1}$, where the temporal context refers to the reconstructed previous frame $\hat{x}_{t-1}$. All baseline methods and RangeCM follow this setting. Since the scanning frequency of the LiDAR sensor is generally fast ($\geq 10$ FPS) [1], the corresponding delay is still acceptable.
>
> On the other hand, RangeCM also supports compressing the partial point cloud. We conduct an experiment where RangeCM first compresses the left half of the range image, and then encodes the right half. This coding pipeline maintains almost the same compression performance, where the BD-Rate increase is **0.24%** and **0.18%** for geometry and intensity compression, respectively. The encoding and decoding times of partial scan are shown in the following table. We have included these results in **Appendix D.9**.
>
> |   Data   | Encoding Time | Decoding Time |
> |:--------:|:-------------:|:-------------:|
> | Complete Scan |     0.22s      |      0.27s     |
> |  Partial Scan|     0.14s      |      0.16s     |
>
> **W2 and Q2: Decoding Error Accumulation**
>
> RangeCM exploits the **reconstructed** previous scan ${\hat{x}}_{t-1}$ to predict the current frame, rather than the ground-truth previous scan. We have clarified this setting in Section 4.3. The proposed comprehensive context model is a lossless compressor, and the decoding error is only introduced by quantization. Therefore, the decoding error of the reference frame does not influence the reconstruction quality of the current frame, and the decoding error does not propagate through the inter-frame prediction.
>
> **References**
>
> [1] A High-Definition LIDAR System based on Two-Mirror Deflection Scanners. IEEE Sensors Journal.

---

> ### Author Response · Authors · 2025-11-20
> **Response to Reviewer uEV9 - 2**
>
> **W3 and Q4: Hardware configurations**
>
> Since most baselines do not release the code, we directly quote the coding latency results from their papers. Here, we clarify the hardware settings of different methods. RangeCM is evaluated on the A6000 GPU, EHEM and RIDDLE are tested with the V100 GPU, while Unicorn and RENO are evaluated with the RTX 3090 GPU. The computing performance of these devices is basically comparable. The Floating Point Operations Per Second (FLOPS) for FP32 data and memory bandwidth (GB per second) of these devices are shown below [2,3]:
>
> |     　    | RTX 3080 | V100 | RTX 3090 | RTX A6000 |
> |:---------:|:--------:|:----:|:--------:|:---------:|
> |   FLOPS   |   59.5   |  112 |    142   |   154.8   |
> | Bandwidth |    760   |  900 |    936   |    768    |
>
> This table suggests that although the evaluation hardware is different, the comparison on coding latency is basically fair. To further demonstrate the low-latency coding strength of RangeCM, we additionally test its coding speed on the RTX 3080 GPU, which is less powerful than all the above listed devices. The results below imply that RangeCM preserves faster coding speed on this entry-level GPU. Please refer to Appendix D.4 for a more detailed comparison.
>
> **Coding times (in seconds) for geometry compression**
>
> |    Method   | Encoding Time (SemanticKITTI) | Decoding Time (SemanticKITTI) | Encoding Time (WOD) | Decoding Time (WOD) |
> |:-----------:|:-----------------------------:|:-----------------------------:|:-------------------:|:-------------------:|
> |    G-PCC    |              0.95             |              0.48             |         1.24        |         0.62        |
> |     RENO    |              0.07             |               -               |          -          |          -          |
> |   Unicorn   |              2.83             |              2.50             |          -          |          -          |
> |    RICNet   |              0.63             |              0.43             |          -          |          -          |
> |    RIDDLE   |               -               |               -               |         0.53        |         0.97        |
> |  RangeCM-G  |              0.09             |              0.09             |         0.21        |         0.23        |
> | RangeCM-GI  |              0.09             |              0.09             |         0.21        |         0.24        |
>
> **Coding times (in seconds) for intensity compression**
>
> |   Method   | Encoding Time (SemanticKITTI) | Decoding Time (SemanticKITTI) | Encoding Time (WOD) | Decoding Time (WOD) |
> |:----------:|:-----------------------------:|:-----------------------------:|:-------------------:|:-------------------:|
> |    G-PCC   |              0.84             |              0.75             |         0.59        |         0.65        |
> | RangeCM-GI |              0.12             |              0.12             |         0.24        |         0.27        |
>
> **W4: Reproducibility**
>
> Code and model weights will be released upon the acceptance of the paper. Meanwhile, we have provided sufficient details about the model architecture and experiment settings in Appendix A and Appendix B to facilitate reproduction.
>
> **References**
>
> [2] NVIDIA V100 Datasheet. https://images.nvidia.com/content/technologies/volta/pdf/volta-v100-datasheet-update-us-1165301-r5.pdf
>
> [3] NVIDIA Ampere GA102 GPU Architecture Whitepaper. https://www.nvidia.com/content/PDF/nvidia-ampere-ga-102-gpu-architecture-whitepaper-v2.pdf

---

> ### Author Response · Authors · 2025-11-20
> **Response to Reviewer uEV9 - 3**
>
> **Q3: Comparison against other baselines**
>
> The original paper did not compare RangeCM with MuSCLE and BIRD-PCC because they do not outperform the selected baselines, such as Unicorn and RIDDLE. RangeCM is additionally compared with MuSCLE in **Appendix D.10**. RangeCM-GI achieves a BD-Rate improvement of **-49.69\%** compared to MuSCLE on the SemanticKITTI dataset, which verifies its superiority.
>
> BIRD-PCC provided neither the source code nor the results on SemanticKITTI or WOD. It uses SemanticKITTI as the training set, and evaluates the model on the KITTI-360 dataset. Therefore, it is difficult to perform an accurate comparison. Here, we present a rough comparison. Compared to G-PCC, BIRD-PCC delivers a BD-Rate gain of **-13.55\%**, while RangeCM-GI achieves **-56.51\%** BD-Rate improvements. It demonstrates that RangeCM significantly outperforms BIRD-PCC.
>
> **Q5: Fairness and reliability of the comparison**
>
> Most baseline methods do not provide the source code, so we quote the performance of baseline methods from their papers, and leave the entries blank if the baseline paper did not provide the corresponding results. The comparison is fair and reliable because RangeCM and baseline methods follow the same experiment settings (e.g., training-testing dataset partition and evaluation metric settings).
>
> In Table 1, we test RangeCM without using the camera context to promise the same dataset split. As demonstrated in **Appendix D1**, using camera context can further improve (rather than degrade) RangeCM’s performance. This experiment indicates that we compare a relatively weak RangeCM with the baseline methods on the KITTI dataset, and even this model still surpasses all baselines. Therefore, the comparison in Table 1 fairly demonstrates the advantages of RangeCM.
>
> **Q6: Advantage over G-PCC**
>
> As shown in Table 1, the coding speed of G-PCC is much slower than RangeCM. Therefore, RangeCM provides significant advantages over G-PCC in latency-sensitive applications such as low-latency communication. Regarding the generality, the GPU devices have been widely used in autonomous driving vehicles [4], because various powerful learning-based perception methods also rely on the GPU. Therefore, the generality is not a severe problem, and RangeCM still has wide application scenarios. Finally, we would like to emphasize that the compression ratio is also important for practical applications. For example, the communication bandwidth is still scarce in many tasks such as cooperative perception. There exists a trade-off between bandwidth and performance of the cooperative perception algorithms [5]. In this context, an effective compressor may improve the perception performance by saving the bandwidth. Moreover, a high compression ratio helps save storage space. For instance, if the user hopes to save the scanned LiDAR point clouds for data analysis or monitoring, an effective compressor may store more scans under the same storage capacity limitation.
>
> **Q7: In-Vehicle Deployment**
>
> We evaluate RangeCM on the RTX A6000 GPU (48GB Memory) and RTX 3080 GPU (10GB Memory), which are not particularly designed for autonomous driving applications. However, please note that the practical in-vehicle GPU platforms have comparable or even stronger computing performance. For example, the Nvidia Driver AGX Thor platform offers **1000 Tera Operations Per Second (TOPS)** for INT8 data [6]. In contrast, the RTX A6000 and RTX 3080 offer only **619.4 and 476 TOPS** [3]. Furthermore, the AGX Thor is built on the latest Blackwell architecture, while the RTX A6000 and RTX 3080 adopt the previous-generation Ampere architecture. Besides, the peak memory consumption of RangeCM is only **3GB**, while AGX Thor has 64GB system memory (shared by CPU and GPU), which is sufficient to deploy our model. Finally, the AGX Thor integrates hardware-level optimized I/O interfaces for camera and LiDAR sensors, which enables fast I/O as well. Therefore, we believe RangeCM can preserve low-latency processing speed in practical systems. We have added these discussions in **Appendix E**.
>
> **Q8: Other problems**
>
> We highly appreciate your kind advice. We have revised the paper accordingly.
>
> We are pleased to provide further explanations if you have any remaining questions.
>
> **References**
>
> [4] NVIDIA DRIVE Partner Ecosystem. https://www.nvidia.com/en-us/solutions/autonomous-vehicles/partners/
>
> [5] UMC: A Unified Bandwidth-efficient and Multi-resolution based Collaborative Perception Framework. ICCV 2023.
>
> [6] NVIDIA DRIVE AGX Thor Development Platform. https://developer.download.nvidia.com/drive/docs/nvidia-drive-agx-thor-platform-for-developers.pdf

---

> > ### Comment · Reviewer_uEV9 · 2025-11-27
> >
> > Thank you for the detailed response, since my concerns have been addressed, I would like to raise my score.

---

> > > ### Author Response · Authors · 2025-11-27
> > >
> > > We sincerely appreciate your kind feedback. We are deeply grateful for your thorough review and insightful suggestions, which have been instrumental in improving the quality of this work.

---

### Official Review · Reviewer_9wkT · 2025-11-05

**Soundness:** 3
**Presentation:** 2
**Contribution:** 3
**Rating:** 6
**Confidence:** 1

**Summary:**

This paper presents a neural LiDAR point cloud compression method based on 2D context modeling, addressing the trade-off between compression efficiency and coding speed that limits existing 3D context-based methods such as voxel or octree approaches. By leveraging a 2D context structure, the proposed method achieves fast and efficient compression while supporting joint geometry and intensity compression. Experimental results show substantial gains in both rate-distortion performance and coding speed.
The contributions of the paper can be summarized as follows: 1. Develop a new paradigm for low-latency LiDAR compression. 2. Propose a comprehensive context model that integrates spatial, temporal, and camera features for LiDAR compression. 3. Design a joint compression backbone that predicts LiDAR geometry and intensity based on a hybrid context.

**Strengths:**

1. The method integrates multi-scale spatial context, flow-based temporal context, and deformable-attention camera context, with context refinement addressing causality and alignment issues. Moreover, the hybrid context jointly supports geometry and intensity compression, reducing redundant computation and significantly improving practical efficiency compared to prior methods such as Unicorn.

2. The paper presents thorough evaluations, including different LiDAR resolutions, coding latency, and impact on downstream tasks like PointPillars, demonstrating robustness and transferability. Ablation studies clearly quantify the contributions of camera, temporal, and multi-scale contexts, showing the effectiveness of each component.

**Weaknesses:**

1. It is unclear whether the VAE used for compressing range-view optical flow was specifically trained on flow data. Without training on range-view optical flow, the VAE may not be able to effectively encode the unique spatial patterns and dynamics of LiDAR flow, potentially leading to high reconstruction error and suboptimal bit allocation. The authors are encouraged to clarify whether the VAE was trained on optical flow and, if not, to discuss the potential impact on compression performance.

2. Material properties and reflections do not directly correspond to visual appearance, which limits the effectiveness of using deformable attention to predict intensity from camera images. The authors are encouraged to discuss whether incorporating more physics- or sensor-based priors (e.g., material classification combined with simplified BRDF models) could improve intensity prediction, or to analyze the potential upper bound achievable by purely data-driven approaches.

**Questions:**

1. Training and hyperparameter details: Please provide a complete description of the training hyperparameters for RangeCM-G and RangeCM-GI (e.g., batch size, optimizer, learning rate schedule, and total training steps). If any baseline methods were re-implemented, please clarify the details of their re-training procedure.

2. Adaptation to different scanning patterns: The appendix presents evaluations on 32-/128-line LiDARs (Tables 6 and 7). Could the authors clarify the robustness of the range image mapping when the sensor’s emission pattern differs from that during training (e.g., different elevation angles)? Would retraining be necessary for each sensor, or could fine-tuning suffice?

3. Limitations and failure cases: In which real-world scenarios do the authors expect RangeCM’s performance to degrade significantly (e.g., extreme weather, reflections from water or snow)? Could some failure cases and analyses be provided to illustrate the system’s limitations?

---

> ### Author Response · Authors · 2025-11-20
> **Response to Reviewer 9wkT - 1**
>
> We sincerely appreciate the reviewer for the thorough review and valuable suggestions. We hope the following responses can address your concerns.
>
> **W1: Whether VAE is trained on flow data**
>
> The VAE for optical flow compression is not trained on the range-view optical flow data. The entire model, including this VAE, is jointly optimized in an end-to-end manner following the loss function of Eq. 7. The ablation studies in Section 5.3 demonstrate the effectiveness of this training strategy. Furthermore, the visualizations in Fig. 12 demonstrate that the decoded optical flow conveys the motion patterns between two frames.
>
> We agree that the separate training of the VAE could potentially improve the compression performance. However, the multi-stage training also complicates the training pipeline: we have to first train a flow estimation model, then the flow compression VAE, and finally train all modules jointly. It requires considerable engineering efforts to adjust hyperparameters and optimize the training recipe. For example, we have to train the model multiple times to find the appropriate rate-distortion trade-off parameter $\lambda$ for lossy flow compression. Besides, since the best $\lambda$ may vary across bitrate levels, we need to train multiple models for different bitrates. In contrast, the proposed end-to-end training strategy is simple and straightforward. It saves training cost, supports variable-rate compression, and delivers satisfactory performance as well.
>
> **W2: Whether incorporating physical-based prior could improve intensity compression**
>
> Using accurate materials, 3D shape, and the corresponding BRDF, it is probably feasible to enhance intensity compression performance. Specifically, we may first infer the material and shape based on the multi-view camera images through an inverse rendering model, and then predict the reflectance intensity based on the rendering function. However, several challenges still remain in practice. Firstly, most inverse rendering methods work in a per-scene per-train manner, which is prohibitively expensive for low-latency applications. Furthermore, existing feed-forward inverse rendering methods are mostly trained for object-level rather than scene-level prediction [1], and thus they may not perform well in complicated autonomous driving or robotic scenes. Besides, since real-world objects are diverse and complex, it may be difficult to find an accurate BRDF. Finally, other unknown factors, such as environment light and atmospheric scattering, may also influence the prediction accuracy. Due to these challenges, we currently leave these explorations for future work.
>
> **Reference**
>
> [1] LIRM: Large Inverse Rendering Model for Progressive Reconstruction of Shape, Materials and View-dependent Radiance Fields. CVPR 2025.

---

> ### Author Response · Authors · 2025-11-20
> **Response to Reviewer 9wkT - 2**
>
> **Q1: Training and hyperparameter details**
>
> RangeCM-G and RangeCM-GI share the same training settings. They are trained by the AdamW optimizer with a batch size of 8. The models are trained for 2M steps on WOD and 0.7M steps on KITTI. The learning rate is set to 1e-4 initially, which decreases to 5e-5 after 60% training steps. These settings have been clarified in **Appendix B.1**.
>
> As most baseline methods provide neither the source code nor sufficient details for reproduction, we compare with them by quoting their results. Since we strictly follow the same experiment setting (e.g., training/testing set partition and evaluation metrics), the comparison is fair and reliable.
>
> **Q2: Adaptation to different scanning patterns**
>
> To investigate whether RangeCM is robust to scanning patterns, we fine-tune a RangeCM-GI model trained on the WOD to encode the point clouds from the KITTI dataset. Notably, the laser emission angles are different on these two datasets. We fine-tune the model for 9K steps, and the results show that the fine-tuned model achieves comparable compression performance to the one trained on KITTI. The fine-tuned model delivers a minor BD-Rate increase of **3.84%** and **3.25%** for geometry and intensity compression, respectively. It demonstrates the robustness of RangeCM. These results have been included in **Appendix D.8**.
>
> **Q3: Limitations and failure cases**
>
> RangeCM may be less effective on extreme weather (e.g., fog and snow) and dark scenes (e.g., night). In these cases, the camera images provide less informative context, and hence the compression performance may degrade as well. As a failure case, RangeCM costs 24.4% more bitrate than its average performance when compressing the point clouds based on camera images taken in a dark scene, which are visualized in Fig. 11.
>
> The intensity compression performance may decrease when the reflectance properties of real-world objects are complicated. For example, it may be difficult to predict reflectance from woodland, water, and snow due to their complex reflectance patterns. The limitation and failure case have been discussed in **Appendix G**.
>
> We are pleased to provide further explanations if you have any remaining questions.

---

### Author Response · Authors · 2025-11-20
**General Response**

We sincerely thank all reviewers for the detailed and constructive comments. We appreciate that the reviewers have recognized RangeCM’s thoughtful design (Reviewer uEV9), strong compression performance (Reviewers 9wkT, uEV9, and MSXZ), and fast coding speed (Reviewers 9wkT and MSXZ). According to the suggestions from reviewers, we have revised the paper from the following perspectives.

**Supplement experiments and analysis**

1.	The performance of generalizing the pretrained model to different scanning patterns is evaluated in Appendix D.8 (Reviewer 9wkT).

2.	The limitations and failure cases of RangeCM are analyzed in Appendix G (Reviewer 9wkT).

3.	Experimental results for coding partial point clouds are presented in Appendix D.9 (Reviewer uEV9).

4.  We evaluate the detailed coding latency of RangeCM on the RTX 3080 GPU and compare this runtime with baseline methods in Appendix D.4 (Reviewer uEV9).

5.	The comparison between RangeCM and MuSCLE is provided in Appendix D.10 (Reviewer uEV9).

6.	The in-vehicle deployment of RangeCM is discussed in Appendix F (Reviewer uEV9).



**Specify method details**

1.	The training settings of RangeCM are clarified in Appendix B.1 (Reviewer 9wkT).

2.  The specific BD-Rate improvements (i.e., average bitrate saving) over each baseline achieved by RangeCM are presented in Appendix D.11 (Reviewer MSXZ).

**Correct typos**

We have corrected the typos in L152, L198, and Figure 2 (Reviewer uEV9).

These modifications are marked in blue in the revised version. We hope these discussions and responses can address all the concerns, and we are pleased to provide more explanations if the reviewers have any further questions. Finally, we sincerely appreciate the engagement of the reviewers and AC, and we look forward to receiving further feedback from the reviewers.

---

### Author Response · Authors · 2025-11-27
**Kind Invitation for Discussion**

Dear Reviewers,

We sincerely appreciate your efforts and dedication, which have been invaluable in enhancing the quality of this work.

In response to your insightful comments, we have revised the paper to address the raised concerns. As the discussion period is drawing to a close, we would be truly grateful to receive any further feedback or suggestions you may have regarding our revisions and responses. We are also pleased to provide any additional clarification, should any points require further explanation.

Once again, we would like to express our deepest gratitude for your time and consideration.

Best regards,

The Authors

---

### Author Response · Authors · 2025-12-01
**Final Comment**

Dear AC and reviewers,

We are deeply grateful for your time and efforts in the review process, as well as your insightful suggestions and constructive discussions. **We are particularly encouraged that the reviewers had reached a consensus in their positive rating of this paper after discussion**. However, due to the unexpected change of the review policy, we would like to further summarize our explanations regarding the major concerns raised by reviewers.

**1. Motivation of RangeCM**: RangeCM aims to improve the compression latency and rate-distortion performance, rather than reducing the perception runtime. Nevertheless, it also supports partial scan compression, making it compatible with the low-latency early-prediction perception algorithms.

**2. Temporal Error Accumulation**: RangeCM uses the decoded previous scan as the temporal context, rather than the ground-truth previous scan. Therefore, the experiments have already taken the impact of the decoding error into consideration. Furthermore, the decoding error does not propagate through temporal prediction, because the compression distortion only arises from quantization, which is independently applied to each scan.

**3. Comparisons to Baselines**: As most baselines do not release the source code, we directly cite the results reported in their papers. The comparison is fair and reliable because we follow their training/evaluation settings. Regarding inconsistent hardware settings that may affect coding time comparisons, we evaluate RangeCM on a GPU (RTX 3080) which is less powerful than the ones used in baselines. RangeCM still delivers significantly faster decoding time than most baselines under this setting.

**4. Practical Deployment**: In-vehicle edge devices (such as NVIDIA Drive AGX Thor) offer comparable or even better computational capability than the hardware used in our experiment (i.e., RTX A6000 and RTX 3080). Besides, the coding time of RangeCM can be further enhanced by model quantization, pruning, and distillation techniques. Therefore, we believe RangeCM can maintain the low-latency coding performance for in-vehicle deployment.

We hope these explanations and our previous responses may address your concerns. Once again, we hope to express our heartfelt thanks for your dedication and contributions to the review process.

Best regards,

The Authors

---

### Meta-Review · Area_Chair_xVcH · 2026-01-15

**Summary:**

[AC:] The initial rates are 6, 4, 8. Notably, the reviewer with the lowest score decided to increase the rating to 6 after reading the authors' responses. Moreover, one reviewer commented that this is a good paper. I do not see any ill intent in the rebuttal period, and would recommend accepting the paper.

**Reviewer Concerns:**

[AC: Most comments have been addressed properly.]

**Reviewer Scores:**

[AC:] The initial rates are 6, 4, 8. Notably, the reviewer with the lowest score decided to increase the rating to 6 after reading the authors' responses. Moreover, one reviewer commented that this is a good paper. I do not see any ill intent in the rebuttal period, and would recommend accepting the paper.

---

### Decision · Program_Chairs · 2026-01-26

Accept (Poster)